# Accuracy Assessment of Low-Cost Lidar Scanners: An Analysis of the Velodyne HDL–32E and Livox Mid–40's Temporal Stability

**Carter Kelly** [1,2,3,*] , **Benjamin Wilkinson** [2,3] , **Amr Abd-Elrahman** [2,3,4] , **Orlando Cordero** [2,3] and **H. Andrew Lassiter** [3,5]

1 Geospatial Information Science Program, Department of Geography and Environmental Engineering, United States Military Academy (USMA), West Point, NY 10996, USA

2 Geomatics Program, School of Forestry, Fisheries, and Geomatics Sciences, University of Florida, Gainesville, FL 32611, USA

3 Geospatial Modeling and Applications (GMAP) Lab, School of Forestry, Fisheries, and Geomatics Sciences, University of Florida, Gainesville, FL 32611, USA

4 Gulf Coast Research and Education Center, Institute of Food and Agricultural Sciences, University of Florida, Wimauma, FL 33598, USA

5 School of Civil and Construction Engineering, Oregon State University, 101 Kearney Hall, Corvallis, OR 97331, USA

* Correspondence: carter.kelly@westpoint.edu

**Abstract:** Identifying and mitigating sources of measurement error is a critical task in geomatics research and the geospatial industry as a whole. In pursuit of such error, accuracy assessments of lidar data have revealed a range bias in low-cost scanners. This phenomenon is a temporally correlated instability in the lidar scanner where the measured distance between target and sensor changes over time while both are held stationary. This research presents an assessment of two low-cost lidar scanners, the Velodyne® HDL–32E and Livox® Mid–40, in which their temporal stability is analyzed and methods to mitigate systematic error are implemented. By immobilizing each scanner as it observes a stationary target surface over the course of multiple hours, trends in scanner precision are identified. Scanner accuracy is then determined using a terrestrial lidar scanner, the Riegl® VZ-400, to observe both subject scanner and target, and extracting the distances between scanner origin and observed surface. Patterns identified in each scanner's distance measurements indicate temporal autocorrelation, and, by exploiting the high linear correlation between scanner internal temperature and measured distance in the HDL–32E, it is possible to mitigate the resulting error. Application of the proposed solution lowers the Velodyne® scanner's measurement RMSE by over 60%, providing levels of measurement accuracy comparable to more expensive lidar systems.

**Keywords:** lidar; UAS; accuracy assessment; Velodyne HDL–32E; Livox Mid–40; range bias

## 1. Introduction

While accuracy assessment requirements may differ between industries and data applications, the need for understanding a measurement's accuracy is universal; without understanding an instrument's capabilities and limitations, users are limited in their ability to use it to their own ends. A measuring instrument's accuracy is optimized (or restricted) by a variety of factors: the experience and competence of users, the conditions of its use, and the process by which its data is processed all contribute to the overall uncertainty and must be considered when determining the accuracy of the end products.

An accuracy assessment is a deliberate process by which the closeness of a measured or computed value is compared to an accepted significantly higher accuracy (often termed "true") value of a particular quantity across the entire area of interest being observed [1].

This informs the users of the quality of the instrument and the derived product, and underwrites decisions made regarding the data's applications. In order to report the accuracy of geospatial data, accuracy assessments are generated and published in conjunction with the collected data and its associated products.

While they are useful evaluations of an instrument's performance, there are deliberate distinctions between precision and accuracy. Precision is defined as the "closeness with which measurements agree with each other", whether or not the measurements "contain a systematic bias" [1]. Accuracy takes this one step further by comparing collected measurements to the location and orientation of target features within a real-world mapping frame through the use of deliberately surveyed check points or other "truth" data [1]. The accuracy of a dataset informs users of how well it agrees with the "ground truth" and informs appropriate use of the data with other collections or measurements of features within the captured area of interest. Considering stability takes performance evaluation one step further by assessing the instrument's ability "to maintain its metrological properties constant with time" [2].

Parameters for assessing lidar system suitability for applications across different environments include the stability, precision (repeatability), and accuracy of their observations. This paper presents an accuracy assessment of two lidar scanners found in the topographic mapping community, and, using the assessment findings, proposes a methodology for deliberately mitigating error in their distance measurements.

### 1.1. Background

The research presented in this paper was first inspired by the results from Lassiter et al. [3], which sought to present a method to detect and measure both the absolute horizontal and vertical errors of UAS lidar surveys. This study was conducted using the Phoenix Lidar Systems® Scout-32 mobile mapping system, which includes (among other sub-systems) a Velodyne® HDL–32E lidar scanner. Through their analysis, the authors identified that there was a "positive bias in the height dimension" for flights conducted earlier in the study (first hour of flight time), versus those conducted later in the day. The temporal nature of the bias indicated that the scanner itself had a previously unaccounted error source, which further investigation suggested could be "a result of the scanner not being properly warmed up" (ibid.).

This finding is supported by existing literature, where Velodyne® lidar scanners have been found to consistently exhibit temporal instability within the first hour of operation. Glennie et al. [4] found that there were "long term variations in the scanner range" in their Velodyne® HDL-64E S2, though they were unable to attribute this error to one specific source. Chan et al. [5] found that "an approximate warm-up time of 2000 s was found for most lasers", with each individual laser (of the HDL–32E's 32 scanners) exhibiting a unique pattern of instability (range walk) when laser incidence and reflection angles were held constant for all observations. Glennie et al. [6] designed and implemented a methodology for measuring the stability of a low-cost Velodyne® lidar scanner, the VLP-16. Like their previous research, Glennie et al. found evidence of a range walk in their scanners through the duration of their experiments, citing it as the reason "that the geometric calibration of the unit is not consistent" [6]. Less consistent, however, have been the causes to which this instability has been attributed, and minimal research exists where this instability's impact on the absolute accuracy of the scanner's distance measurements is analyzed and presented.

Less established in existing literature is the stability and UAS-mapping utility of the ultra-low cost Risley-prism based laser scanner from Livox®, the Mid–40. First available on the market in 2019 for the cost of just $600, the Mid–40 was initially designed as an automotive lidar solution capable of detecting targets at up to 260 m within its circular scan area [7]. In their analysis of the scanner and the point clouds it produced, Ortiz Arteaga et al. [8] identified a "ripple effect" in close distance, indoor tests when observing a planar object that was found to have a significant "influence on range and angular measurements". Glennie et al. [9] found that "the Livox® Mid–40 sensor performance appears to be in con-

formance with the product specifications, with a ranging accuracy of approximately 2 cm"
but, due to the beam divergence across the scan area, was "unable to properly model sharp
edges and small features such as poles". Studies of a similar Livox® scanner, the Avia, have
identified similar shortcomings. Teppati Losè et al. found that, while integrated as part of
the DJI® Zenmuse L1, the Avia failed to capture "a satisfactory level of detail" for objects
defined by sharp edges (emergency stairs on the side of a building) [10]. Brazeal et al. [11],
as part of the inclusion of the Mid–40 in the OpenMMS project [12], found that while "the
angular observations for the Mid–40 sensor are more accurate than the manufacturer's
specification of <0.1°", there was a systematic error in the "Cartesian coordinate repre-
sentation of the observations" as "a result of the emergent beam not coinciding with the
origin of the sensor's reference coordinate system" [11]. As a result of this error, correc-
tions to the calculated Y and Z coordinates were required by "upwards of 4.5 mm" [11].
While potential sources of systemic error in the Mid–40's observations have been identi-
fied in literature, there has been limited discussion on the impact of these factors on the
scanner's absolute accuracy.

*1.2. Research Questions*

Given that there are a handful of published studies regarding the accuracy and pre-
cision of small form laser scanners, an important question remains: can the stability and
absolute accuracy of distance measurements be directly observed in a robust, repeatable
manner? Based on the demand for rigorous accuracy assessments of low-cost lidar scanners,
the research presented here seeks to answer this question for the benefit of both the members
of academia and the greater laser scanning community. As a whole, this study introduces
methods to assess and increase the absolute accuracy of low-cost, sUAS-mountable laser
scanners, informing their use in applications from rigorous scientific studies to practical
utilizations. More specifically, this study presents:

1.  A methodology to directly observe and analyze the precision and absolute accuracy
    of low-cost lidar scanners;
2.  The identified range walk in low-cost lidar scanners and an analysis of its correlated
    scan parameters;
3.  Measures to correct, adjust out, or otherwise mitigate systematic errors in observations
    for high-accuracy sUAS surveys;
4.  A comparison between like-tier lidar scanners in terms of precision, accuracy, and
    temporal stability.

Through the execution of the outlined methodology, this study shows that low-cost
laser scanners exhibit a temporal distance bias that is highly correlated with internal
scanner temperature, and that survey accuracy can be improved through application of
a per-laser linear regression in multi-laser scanners. Additionally, it is demonstrated that
scan conditions, such as fluctuations in solar radiation on target surface and increased
beam incident angle, have a measurable impact on scanner performance, which in turn can
be effectively mitigated through deliberate selection of survey execution conditions and
flight parameters.

## 2. Materials and Methods

*2.1. Scanners Assessed*

### 2.1.1. Velodyne® HDL–32E

The HDL–32E is a lightweight lidar scanner first introduced by Velodyne® in 2011 as
an "ultra-compact and cost-effective optical sensor for many unmanned robotic vehicles [5].
Since its introduction, it has been popularized in use in the robotics, autonomous vehicle,
and mobile mapping industries, and has been included as a keyhole system for multiple
mobile lidar scanning systems [5,13,14]. For this study, the HDL–32E used was a sub-
component of Phoenix Lidar Systems® Scout-32 (Figure 1A), a pre-made laser scanning

system featuring a navigation system and IMU (providing six degree of freedom inertial sensor measurements), RGB camera, and purpose-built housing for sUAS operations [14].

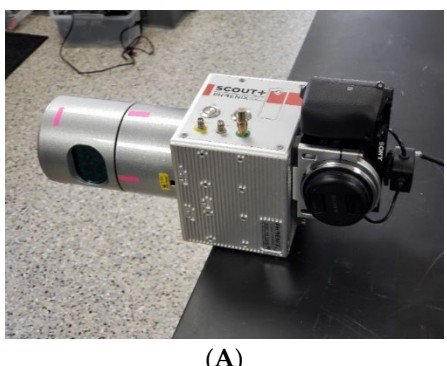
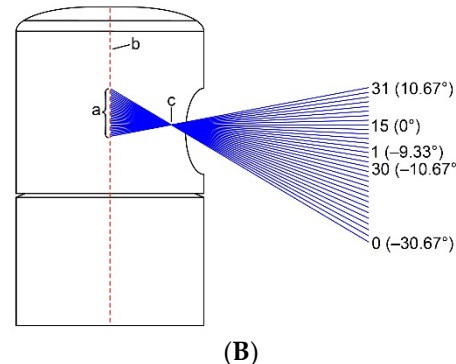

(A)  (B)

**Figure 1.** (**A**) Phoenix Lidar Systems® Scout-32, featuring a Velodyne® HDL–32E Lidar Scanner and Sony® Alpha A6000 24 MP camera, with a (**B**) diagram of Velodyne® HDL–32E, highlighting (a) 32 individual, vertically stacked scanner/receiver pairs that rotate about (b) a central axis oriented through (c) a focal point as they scan a target area. Adapted from [13].

The HDL–32E functions by rotating a vertical array of 32 individual laser emitter/detector pairs about a central axis, with each scanner deliberately aimed with a vertical offset angle creating a fan-like scanning pattern (Figure 1B). As the scanner runs, its continuous spinning allows it to capture 32 distinct scan lines in a 360° ring around its central axis which, when coupled with the scanner's physical motion through 3D space, enables a thorough scan of topographic areas within a moderate distance. The scanner is capable of collecting distance and calibrated reflectivity measurements, rotation angles, and synchronized time stamps (μs resolution) at a rate of up to 695,000 points per second (single return mode) or 1,390,000 points per second (dual return mode) [13]. Further specifications are listed in Table 1.

**Table 1.** Velodyne® HDL–32E Select Specifications [13].

| **Platform** | |
|---|---|
| Overall Dimensions | 24.6 cm × 11.6 cm × 11.6 cm |
| Operating Temperature | −10 °C to +40 °C |
| Weight | 2.40 kg |
| **Sensor** | |
| Laser Wavelength | 903 nm |
| Laser Classification | Class 1 Eye Safe |
| Channels | 32 |
| Measurement Range (min, max) | 1 m, 100 m |
| Measurement Range (resolution) | 0.002 m |
| Manufacturer's Reported Range Accuracy | ±0.02 m |
| Measured Returns | 1 or 2 |
| Field of View (Vertical) | +10.67° to −30.67° (41.33°) |
| Field of View (Horizontal) | 360° |
| Angular Resolution (Vertical) | 1.33° |
| Angular Resolution (Horizontal/Azimuth) | 0.08°–0.33° |
| Beam Divergence | 3 mrad |
| Rotation Rate | 5 Hz to 20 Hz |
| 3D Lidar Data Points Generated: Single/Dual Return Mode | 695,000/1,390,000 points per second |

### 2.1.2. Livox® Mid–40

The Livox® Mid–40 is a recent addition to the lidar scanner industry, introduced by Livox® (a subsidiary of DJI®) in 2019 at a price point significantly below the automotive lidar standard price point ($599 at time of writing) [8]. A small form sensor, the Mid–40 is

advertised by its manufacturer as a low-cost alternative for "autonomous driving, robotics, mapping, security, and other areas from small batch testing to large-scale applications" [7]. The scanner, like the Velodyne® HDL–32E, has been incorporated in to both proprietary and open-source mobile mapping systems available on the open market. For this research however, the scanner was as pictured in Figure 2, separate from any other mapping system component.

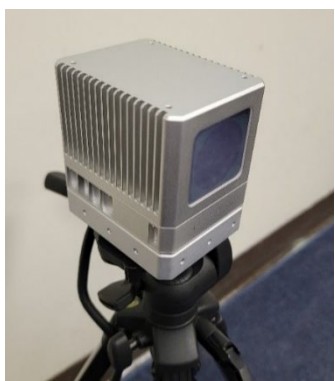

**Figure 2.** Livox® Mid–40 as mounted on a camera tripod.

The Mid–40 itself is capable of collecting up to 100,000 points per second within a circular field of view directly in front of the scanner's glass window. Smaller than 10 cm³, it has been rated to collect points on objects at standoff distances of up to 260 m with precisions of up to 0.02 m (further specifications are listed in Table 2). However, what sets the Mid–40 apart from other low-cost laser scanners is its scanning mechanism. Rather than rotate the laser scanner/receiver pair themselves (as is done in a multi-line scanner like the HDL–32E), the Mid–40 holds the two in a "solid state", immobilized inside the scanner's housing while the outbound and inbound light is directed through a pair of Risley prisms [11]. These wedge-shaped glass lenses act as optical steering mechanisms as they rotate in opposite directions, directing the beam in a rosette-style scan pattern in the circular scan area (Figure 3).

**Table 2.** Livox® Mid–40 Select Specifications [15].

| Platform | |
|---|---|
| Overall Dimensions | 8.8 cm × 7.6 cm × 6.9 cm |
| Operating Temperature | −20 °C to +65 °C |
| Weight | 760 g |
| **Sensor** | |
| Laser Wavelength | 905 nm |
| Laser Classification | Class 1 (IEC 60825-1:2014) eye safe |
| Channels | 1 |
| Measurement Range (min) | 1 m |
| Measurement Range (max) | 90 m at 10% reflectivity |
| | 130 m at 20% reflectivity |
| | 260 m at 80% reflectivity |
| Manufacturer's Reported Measurement Range (precision) (1 σ at 20 m) | 2 cm |
| Beam Divergence | 0.28° (vertical) × 0.03° (horizontal) |
| Measured Returns | 1 |
| Field of View (Circular) | 38.4° |
| Angular Accuracy | <0.1° |
| Point Generation Rate | 100,000 points per second |

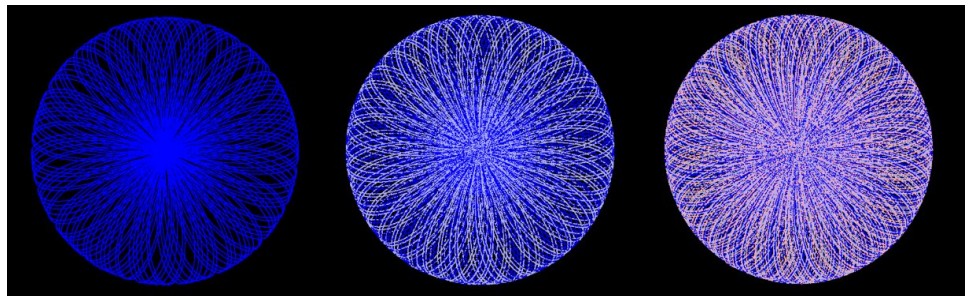

**Figure 3.** Rosette-style scan pattern of the Livox® Mid–40, over a course of three seconds of operation. From left to right, each circle depicts the addition of one-second's worth of data from a stationary scanner observing a stationary target surface.

## 2.2. Research Methodology

### 2.2.1. Experiment Design and Execution

In order to assess the performance (precision, accuracy, and stability) of the Velodyne® HDL–32E and the Livox® Mid–40, it was necessary to identify parameters that may influence the scanner performance over time. In multiple iterations of experiments, the two scanners were immobilized, oriented towards a vertical wall (painted drywall or exposed concrete) between 2 and 30 m away from the scanner, and run for a multiple-hour collection period. A precision assessment was executed by comparing each scanner's measurements of distinct subsections of the stationary target surface over time. Then, using a highly accurate terrestrial lidar scanner (the Riegl® VZ-400), both scanner and target surface were mapped in a dense point cloud, enabling the standoff distance between scanner origin and target surface to be measured for comparison (Note that throughout the rest of this manuscript, these higher-accuracy distances are termed "true/truth" with the understanding that they are appropriate surrogates for actual true values used in accuracy assessment). The comparison of this "true" standoff distance to the measurements recorded by both the HDL–32E and Mid–40 provided an absolute accuracy assessment of each scanner, as well as a detailed understanding of their respective range walks, and how to mitigate any resulting error.

The subject scanners are positioned to maximize the target surface's exposure to emitted lasers. Because the HDL–32E and Mid–40 have dissimilar scan patterns, the two scanners do not share the same specific orientation of each scan axis (X, Y, Z). The HDL–32E, being a line-type scanner, is laid on its side so its Z-axis is horizontal (rather than vertical), and its Y-axis is vertical (rather than horizontal) as shown in Figure 4. The end result of this rotation is that all 32 scan lines are approximately vertical, with a direct relationship between target height and scan angle. For the Mid–40, no rotation is required. As shown in Figure 4, the scanner is maintained in an upright position with the window parallel to the scanning surface.

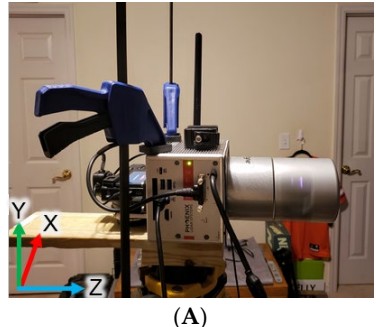

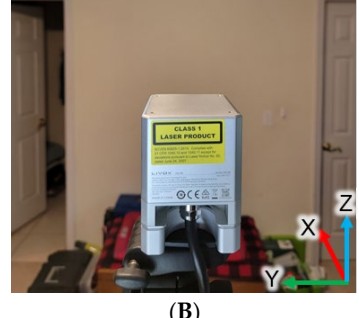

(**A**)   (**B**)

**Figure 4.** Orientation of (**A**) the Velodyne® HDL–32E and (**B**) Livox® Mid–40 during indoor data collection. The scanner orientations shown here were used for all data collection (indoor and outdoor) throughout the duration of this study.

Each experiment was executed in the same fashion: the scanner was immobilized on an adjustable, leveled tripod and oriented approximately perpendicular to the target surface. When conditions were set, the scanner was powered on and, while operating continuously for the entire experiment, observations were recorded for the first two out of every ten seconds. This two-out-of-ten collection method ensured that the lidar point clouds were manageable in size and enabled them to be easily isolated during analysis by timestamp, all while capturing any overall temporal trends in the data. The HDL–32E data were accompanied by a record of scanner diagnostic information, retrieved every 30 s of scanner operation, consisting of scanner internal temperature (°C) and power consumption (V). The Mid–40 lacks the capability to output this information, so no record of its internal temperature or power consumption was available for analysis.

Incident Angle

While both the HDL–32E and Mid–40 record horizontal and vertical scan angles, the incident angle of the scanner's laser beam on the target surface potentially varies between experiments and must be calculated after the conclusion of each study. The incident angle is the difference between the laser's angle of emission and the normal of the target surface, and is calculated using Equation (1), where (X, Y, Z) is the vector between the scanner's origin and a point in the point cloud, and $(N_X, N_Y, N_Z)$ is the normal vector to the target surface. The normal of the target surface is determined by extracting the scanner's observations of the target surface and fitting them to a plane, thereby ensuring both vectors are in the same coordinate system.

$$\text{Incident Angle} = \cos^{-1}\left(\frac{\vec{A} \cdot \vec{B}}{\|\vec{A}\| \cdot \|\vec{A}\|}\right) = \cos^{-1}\left(\frac{(XN_X) + (YN_Y) + (ZN_Z)}{\sqrt{X^2 + Y^2 + Z^2} \times \sqrt{N_X{}^2 + N_Y{}^2 + N_Z{}^2}}\right) \qquad (1)$$

2.2.2. Phase 1: Indoor Experiments (Precision)
Study Site

Indoor experiments, while not necessarily able to replicate the conditions of sUAS topographic surveys in both distances between scanner and target and environmental conditions, enabled a baseline of performance and stability of the laser scanners over time. Additionally, their repeatability allowed the identification of trends across datasets, reinforcing findings of scanner precision (or lack thereof) and associated stability. For these experiments, a temperature-controlled office with 2-m standoff distance from a blank, painted drywall wall was used. Maintaining a room temperature of approximately 22 °C and no sunlight exposure, the scanners were oriented towards the target wall, leveled, and immobilized (Figure 4) and run for three uninterrupted hours. This phase of the study consisted of a total of four separate experiments, two per scanner.

In order to ensure the scanners only collected measurements from uniform sections of the target surface, the HDL–32E was restricted to scan azimuths ($\alpha$) $\pm8.00°$ relative to the target surface normal for all 32 individual laser channels. The Mid–40 required no such restriction so long as it was carefully aimed to collect data from empty portions of the target surface. Because the target surface was a generally planar wall in a controlled environment, observations from each scanner at like scan angles should have been consistent in measured distance, intensity, and incident angle over time. Deviations in any of these measurements indicated a lack (or loss) of scanner stability.

HDL–32E

After the completion of each experiment iteration, the two-second observations from the HDL–32E (Figure 5) were consolidated for analysis. This process was accomplished using Phoenix Lidar Systems® *Spatial Explorer* [16] and Kitware® *VeloView* [17]. Each observation file wassplit by individual channel (channel 00 through 31), from which measured distances were averaged to create a mean observed distance per channel.

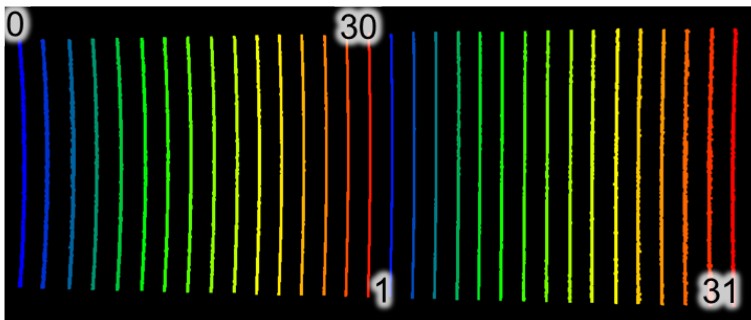

**Figure 5.** Velodyne® HDL–32E raw observations as points representing scan lines from all 32 individual channels (even numbers 0 through 30 from left to right, followed by odd numbers 1 through 31 from left to right).

By repeating this process for all observation files, a list of average distances per channel at each two-second epoch was generated. This simplified and isolated systematic phenomena for each individual channel. These lists were, in turn, consolidated and used to generate basic summary statistics (mean, maximum, minimum, range, standard deviation) for measured distances for each laser and, when combined, for the scanner as a whole (Figure 6).

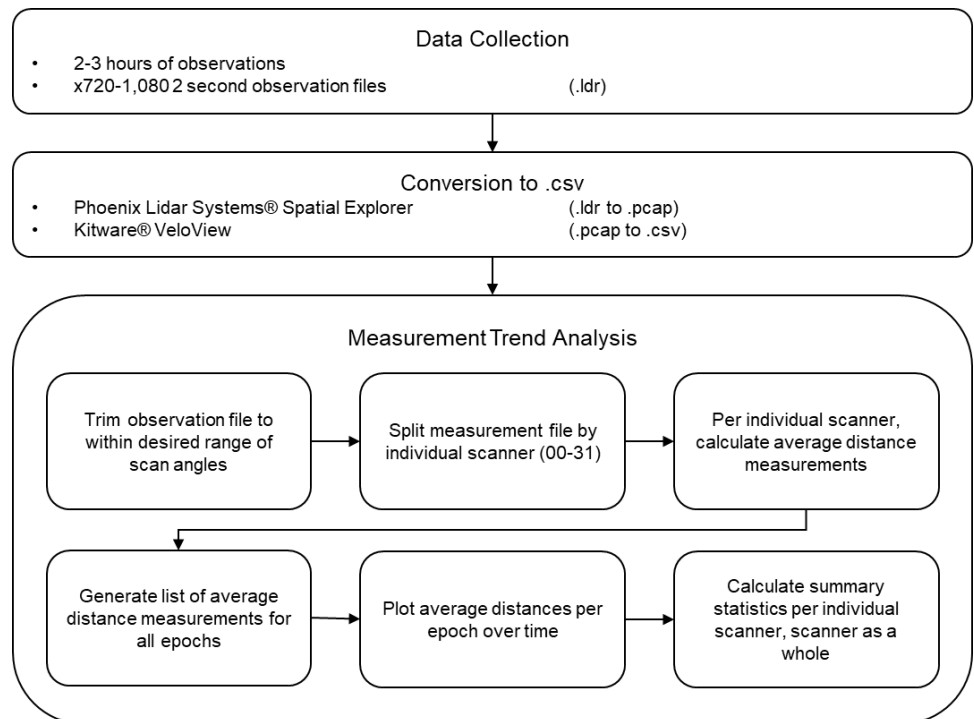

**Figure 6.** Velodyne® HDL – 32E measurement analysis workflow.

Diagnostic data collected for the HDL–32E was downloaded and processed in accordance with the guidance published in the Velodyne® HDL–32E User's Manual, which contains step-by-step instructions for interpretation of the raw data provided by the scanner [13]. Once converted to °C, the measured internal temperatures could be compared, by timestamp, to the HDL–32E's distance measurements. If a suspected correlation existed between any observed parameter and the scanner's observations, the correlation of determination ($R^2$) could be calculated, which would indicate the amount of variation in measured distance that could be explained by the scanner's internal temperature or power consumption.

Mid–40

The raw observations from the Livox® Mid–40 were recorded in a .lvx file format, which is a proprietary file type used by Livox®. As with the HDL–32E, observations from every two-second epoch were saved and converted to a .csv file for analysis. Unlike the HDL–32E, this process could be done using just the Livox® software package Livox Viewer, "convert to .csv" function [18].

Because the Mid–40 only has one scanner/receiver pair (unlike the 32 in the HDL–32E) and due to its rosette-style scan pattern, its observation files cannot be easily divided by individual scan lines. Instead, the observations were simplified for analysis by dividing the scan area into an array using the range of observed Y and Z coordinates (Figure 7). The distance measurements of each point within each cell were then averaged, and this calculated mean distance was assigned as each cell's value. This process was repeated for every observation file, and the resulting series of arrays were combined to generate summary statistics of the scanner's observation over the course of the three-hour experiment.

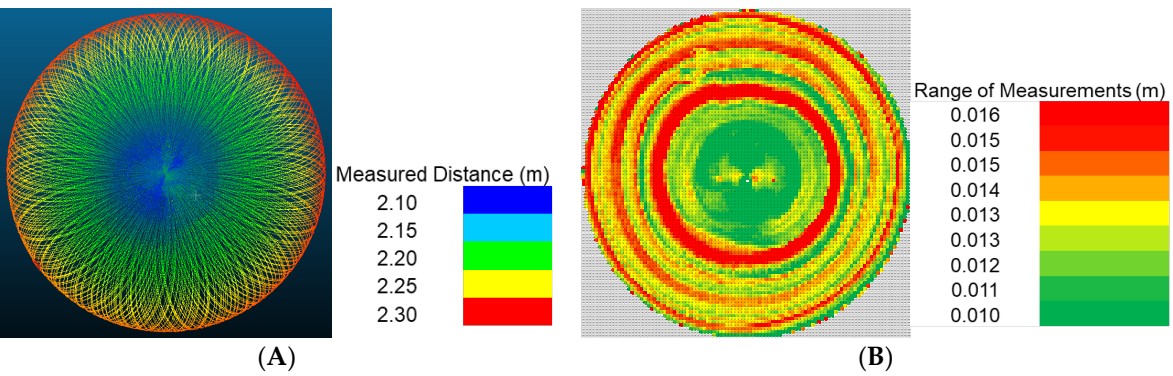

**(A)**                                         **(B)**

**Figure 7.** Breakdown of example Mid–40 data from (**A**) raw observations, colorized by measured distance, to (**B**) a $100 \times 100$ array of cells colorized by range of average distance measurements per cell.

2.2.3. Phase 2: Outdoor Experiments (Precision and Accuracy)

Study Site

Outdoor experiments, used to more closely replicate the conditions of sUAS surveys, were conducted on the permanent parking infrastructure on the University of Florida campus. In order to maintain uninterrupted observation of a relatively flat target surface, the exposed concrete side of one of the university's parking garages was used as the target surface, which was clearly visible from the top of the neighboring (shorter) garage (Figure 8). In this phase, a series of independent experiments were conducted per scanner, 30 m from the target surface during both night and day conditions. While a total of nine separate experiments were conducted under a variety of distances and experiment conditions, four were used for this analysis due to the similar collection parameters between scanners and iterations (Table 3).

**Table 3.** Experiment List and Conditions.

| Lidar Scanner | Standoff Distance | Day/Night | Experiment Date |
| --- | --- | --- | --- |
| HDL–32E | 30 m | Day | 4 June 2021 |
| HDL–32E | 30 m | Night | 9 July 2021 |
| HDL–32E | 30 m | Day | 13 July 2021 |
| HDL–32E | 60 m | Night | 15 July 2021 |
| Mid–40 | 30 m | Night | 4 November 2021 |
| HDL–32E | 30 m | Night | 21 November 2021 * |
| Mid–40 | 30 m | Night | 21 November 2021 * |
| HDL–32E | 30 m | Day | 21 November 2021 * |
| Mid–40 | 30 m | Day | 21 November 2021 * |

* Experiment used in final accuracy assessment.

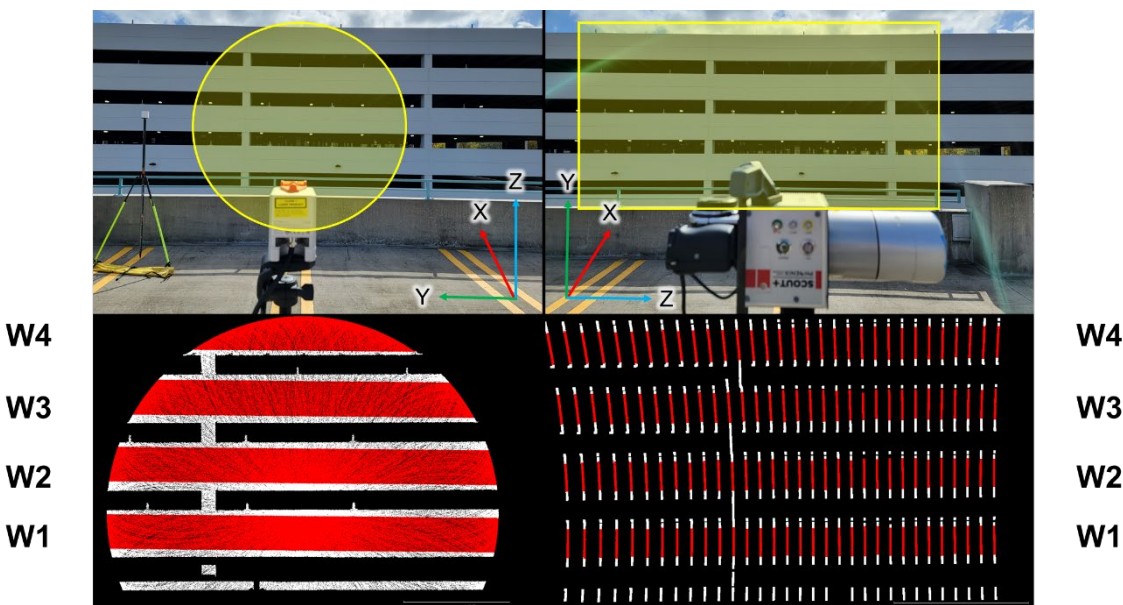

**Figure 8.** Study site and target surface at the University of Florida. Top: The scanners were oriented at the side of an adjacent parking garage (yellow) at standoff distances to reflect a sUAS topographic data collection. Bottom: Scanner observations relative to each scanner are shown with extracted subsections highlighted in red. In order to facilitate reporting of results, the subsections of the target surface are designated W1 through W4, with W1 being the lowest and W4 being the highest observed by the scanners.

By setting the scanners approximately 30 m away from the target wall, four distinct horizontal surfaces were visible in each scan (Figure 8). This separation between sections allowed a comparison of like features between the two scanners, while also enabling the deliberate placement of targets across various scan angles (as described later in this section). Like the indoor experiments, scan azimuths for the HDL–32E were restricted to those that capture the target for extraction and later analysis. A similar restriction was also applied to the Mid–40's measurements as its field of view included more than the desired target surface.

As previously described, there is a known degree of error associated with edge effects while using scanners such as the Mid–40 and HDL–32E [8,9,13,19]. Therefore, after collection was complete, edges of the target surface subsections were cropped out of the observations prior to any assessments. (These sections were easily isolated by using the scan angles of each point as recorded by their respective scanner). Additionally, for all outdoor experiments, a detailed weather record was downloaded from WeatherSTEM portal for Alachua County, Florida [20], which actively mines and stores meteorological data from the University of Florida weather stations at Ben Hill Griffin Stadium and Weimer Hall, approximately 1 km from the study site. The data used for this study included air temperature, atmospheric pressure, and solar radiation, all measured and recorded every 60 s.

The design of the outdoor experiments also included the placement of alignment targets along the target surface (Figure 9) and the inclusion of a Riegl® VZ-400 Terrestrial Lidar Scanner (TLS), which, according to the manufacturer, has a distance measurement accuracy of ±0.005 m and a precision of ±0.003 m. The VZ-400 scanner was used to capture the target surface and assessed Velodyne® or Livox® scanner in a highly accurate and dense point cloud. By comparing measurements from the HDL–32E and Mid–40 to an independent and more accurate source (e.g., those from the VZ-400), this method of assessing data accuracy met the intent of the ASPRS Positional Accuracy Standards [1]. Similar methods of capturing truth data for scanner accuracy assessments can be found in the existing literature, such as [9].

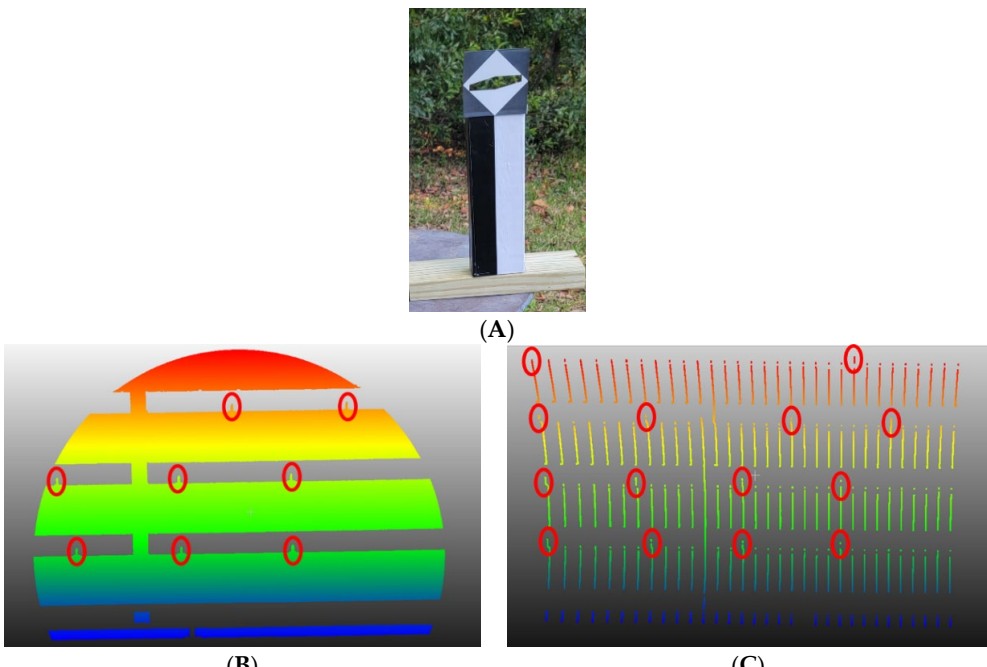

**Figure 9.** (**A**) Alignment targets used for absolute accuracy assessment and their locations as observed with the (**B**) Livox® Mid–40 and (**C**) Velodyne® HDL–32E are shown in selections from each scanner's point clouds.

During the HDL–32E data collection, alignment targets (Figure 9) were placed along the top of each target wall in line with the vertical scan lines being generated by the scanner. Alignment was made possible through use of an IR camera, which was used to detect the flash of individual laser pulses and, in turn, ensured the scan lines impacted on the desired targets. As the targets were placed outside of the identified sections of the walls that were being extracted for analysis, their placement and alignment had no impact on the collected measurements (Figure 9). As the HDL–32E and Mid–40 collections were conducted sequentially to ensure collection conditions were similar, the same alignment targets were used for both scanners despite their independent workflows.

Following the conclusion of each scanner's collection window, the VZ-400 was used to capture the scanner and target surface in the same reference frame in a highly dense and accurate point cloud (Figure 10). These point clouds were used to estimate the "true" distance between each scanner's origin and their observed points, which, when compared to the recorded Velodyne® or Livox® scanner observations, enabled the absolute accuracy assessment of each target scanner.

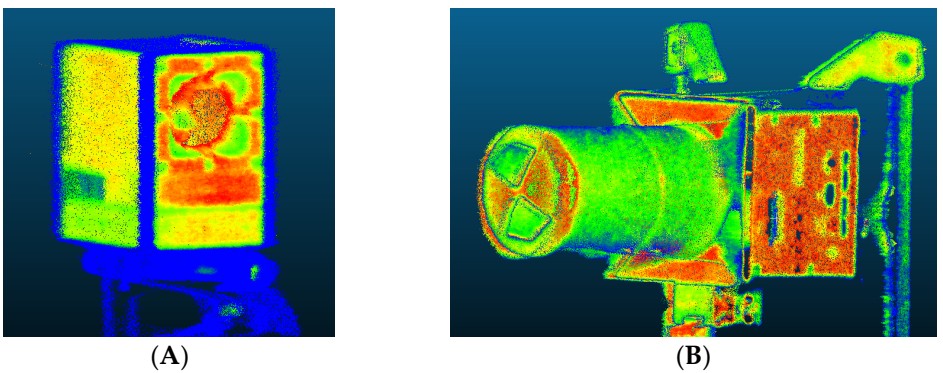

**Figure 10.** Point clouds generated by the Riegl® VZ-400, representing the (**A**) Livox® Mid–40 and the (**B**) Velodyne® HDL–32E.

HDL–32E

The observations generated by the HDL–32E were subject to the same workflow as described in Section 2.1.2. with the additional step of separating the target surface by planar subsection. The VZ-400 data were added to the process through the mensuration of the HDL–32E's scanner origins and focal points (Figure 11) in the VZ-400's point cloud and the projection of its scan lines to extract the truth data for measured distances. Prior to data collection, black and white targets were attached to the top and base of the HDL–32E's cylinder shape, centered as to mark the top and bottom of the central axis shown in Figure 11. These two points were identified in CloudCompare® and used as inputs for the scan line projection script [21].

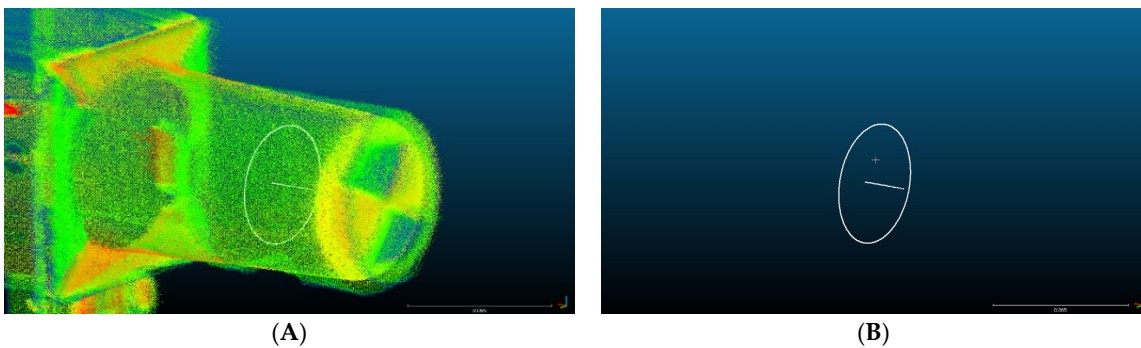

(A)          (B)

**Figure 11.** Velodyne® HDL–32E with individual scanner and receiver pairs (32 total) plotted along central axis with focal point locations plotted for every 0.01° of rotation (**A**) with and (**B**) without the scanner point cloud.

Using the published dimensions of the HDL–32E [13], the locations of all 32 individual diode/receiver pairs were calculated and plotted within the scanner's cylinder. Additionally, the focal point for all diodes (appearing as a white circle shown in Figure 1B) was plotted as it rotates around the central axis for every 0.01° of rotation (Figure 11).

Points representing the scan lines for each individual laser were then plotted along vectors originating at the laser's origin out to the approximate distance of the target surface (Figure 12). Where these vectors intersected the target surface (as scanned by the VZ-400) they were interpolated as an additional scalar field for the target surface point cloud with a width approximately equal to that of the laser's footprint at scanner distance. Subsections of the target surface were cropped to match the HDL–32E observations, and distances between each observed point and scanner origin were used as the "true" distance between scanner and target. The accuracy assessment was then conducted by comparing the "true" distance (as determined by the VZ-400) and measured distance (as collected by the HDL–32E) for each of the four subsections of the target surface.

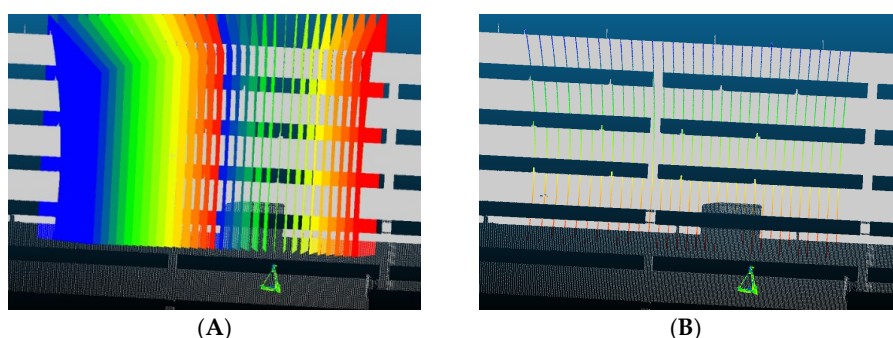

(A)          (B)

**Figure 12.** Projection of Velodyne® HDL–32E scan lines in (**A**) the Riegl® VZ-400 point clouds and (**B**) their interpolation as an additional scalar field on the target surface.

Mid–40

Like the HDL–32E, the workflow for processing observations made by the Mid–40 in this phase was identical to that from the precision assessments of phase two. However, additional steps were required to align and extract the truth data for the absolute accuracy assessment. First, as per the recommendations of [22], each of the Mid–40's observation files were cropped by its measured intensity and radius values to eliminate erroneous measurements and isolate the target surface. The resulting target surface point cloud was aligned with the VZ-400 target surface point cloud using the alignment targets at CloudCompare's® "align two entities by picking equivalent points" tool (Figure 13) [21]. Following alignment, the Mid–40's field of view was interpolated as additional scalar fields in to the VZ-400's point cloud, identifying where the Mid–40's laser hit the target surface. Finally, the distance between each identified point and the Mid–40's origin was calculated using the scans of the Mid–40 in the VZ-400's reference frame, which in turn acted as the truth value for the absolute accuracy assessment.

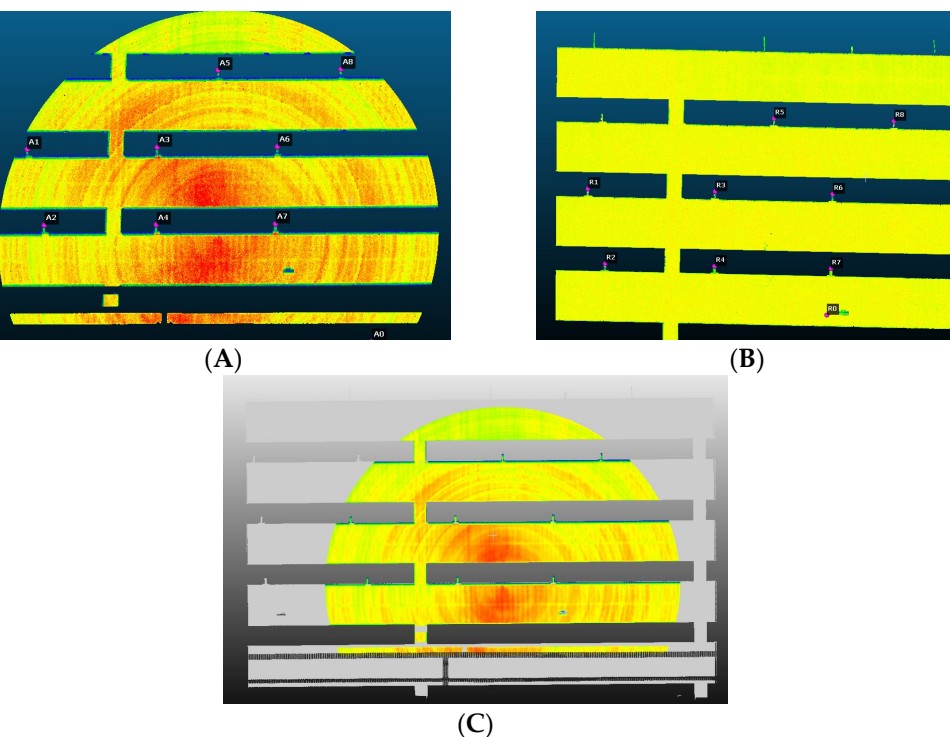

(A)　　　　　　　　　　　　　　　　　　　(B)

(C)

**Figure 13.** Alignment of (**A**) the Livox® Mid–40-point cloud with (**B**) the Riegl® VZ-400 point cloud using alignment targets placed on the target surface and (**C**) subsequent interpolation of scalar fields.

## 3. Results and Analysis

### 3.1. Velodyne® HDL–32E

3.1.1. Phase 1: Indoor Experiments (Precision)

To assess the performance of the HDL–32E as a whole, its measurements of the target surface ($\pm 8°$) from all channels were averaged per epoch (2 out of every 10 s for a 3-h experiment duration) for a series of approximately 1080 average measurements. This series of means was plotted over time (Figure 14) to visualize a number of temporal trends in scanner performance. First, there was a distinct decrease in measured distance over time, most noticeably in the first 30 min of scanner operation; in both iterations of indoor experiments, distance measurements decrease approximately 0.014 m at approximately the same rate over the same period of time. Second, during this initial period of operation, the scanner exhibited a higher standard deviation in measurements–an order of magnitude (or more) higher than in successive 30-min increments, due almost entirely to the observed

temporal bias. Finally, after the initial range walk, there was minimal variation in the overall measured distance; scanner stability appears to increase over time.

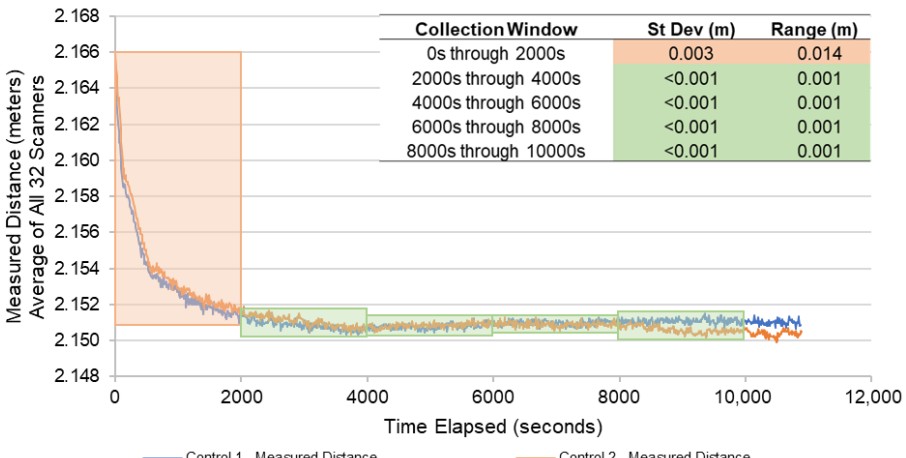

**Figure 14.** Velodyne® HDL–32E average distance measurements over time (average of all points from all 32 scanners within the −8° to +8° scan window) with variations in scanner precision highlighted.

These overall average distance measurements can be broken down by individual scanner, and in doing so it is possible to plot the observations of each of the 32 channels within the HDL–32E over time and compare their stability at an individual level. While all 32 exhibited similar curves in their measurements to the overall plot shown in Figure 14, there was a degree of discrepancy in some of the plots compared to others (Figure 15). To quantify this difference, the range and standard deviation of measured distances over the three-hour experiment for each individual scanner are shown in Table 4.

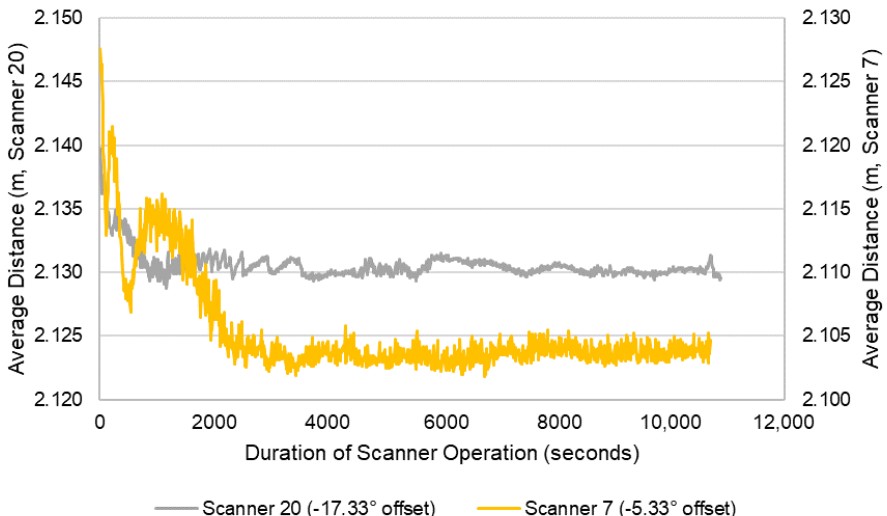

**Figure 15.** Comparison of individual channels' average measured distances for the Velodyne® HDL–32E. The two pairs shown are the ones with the highest (channel 20) and lowest (channel 7) precision in their averaged measurements over time.

The distribution of scanners with relatively high and low measurement range or standard deviation did not appear to be correlated with position along central axis (and resulting offset angle). Variations in measured range were only as high as approximately 0.02 m between the most- and least-precise channels, but this, along with the minimal range in standard deviations did not appear to be statistically significant. As these results were consistent across all three indoor experiments, no one individual channel appeared to negatively influence the HDL–32E's overall precision to a significant degree.

**Table 4.** Summary of Velodyne® HDL–32E indoor experiment indoor measurements (average of all points from each scanner within the −8° to +8° scan window).

| Scanner | Offset | Range (m) | St Dev (m) | Average (m) |
|---|---|---|---|---|
| 31 | 10.67° | 0.011 | 0.001 | 2.226 |
| 29 | 9.33° | 0.013 | 0.001 | 2.217 |
| 27 | 8.00° | 0.011 | 0.002 | 2.196 |
| 25 | 6.67° | 0.016 | 0.001 | 2.180 |
| 23 | 5.33° | 0.012 | 0.002 | 2.172 |
| 21 | 4.00° | 0.018 | 0.002 | 2.162 |
| 19 | 2.67° | 0.016 | 0.002 | 2.143 |
| 17 | 1.33° | 0.024 | 0.003 | 2.158 |
| 15 | 0.00° | 0.021 | 0.002 | 2.124 |
| 13 | −1.33° | 0.015 | 0.001 | 2.119 |
| 11 | −2.67° | 0.015 | 0.001 | 2.111 |
| 9 | −4.00° | 0.018 | 0.002 | 2.107 |
| 7 | −5.33° | 0.026 | 0.004 | 2.105 |
| 5 | −6.67° | 0.021 | 0.002 | 2.101 |
| 3 | −8.00° | 0.027 | 0.003 | 2.099 |
| 1 | −9.33° | 0.014 | 0.001 | 2.093 |
| 30 | −10.67° | 0.017 | 0.002 | 2.103 |
| 28 | −12.00° | 0.019 | 0.002 | 2.105 |
| 26 | −13.33° | 0.013 | 0.002 | 2.105 |
| 24 | −14.67° | 0.019 | 0.002 | 2.114 |
| 22 | −16.00° | 0.019 | 0.002 | 2.120 |
| 20 | −17.33° | 0.011 | 0.001 | 2.131 |
| 18 | −18.67° | 0.017 | 0.002 | 2.128 |
| 16 | −20.00° | 0.013 | 0.002 | 2.144 |
| 14 | −21.33° | 0.022 | 0.003 | 2.150 |
| 12 | −22.67° | 0.021 | 0.003 | 2.160 |
| 10 | −24.00° | 0.018 | 0.002 | 2.168 |
| 8 | −25.33° | 0.015 | 0.002 | 2.192 |
| 6 | −26.67° | 0.017 | 0.002 | 2.207 |
| 4 | −28.00° | 0.010 | 0.001 | 2.225 |
| 2 | −29.33° | 0.015 | 0.002 | 2.250 |
| 0 | −30.67° | 0.012 | 0.001 | 2.253 |
| | Average | 0.017 | 0.002 | 2.151 |

Due to the controlled nature of the indoor experiments, one of the few parameters that changed over the course of the experiment was the internal temperature of the HDL–32E itself. This data, extracted from the self-diagnostic output of the instrument, was converted to °C and plotted, over time, with the overall average measured distances from Figure 14 (Figure 16).

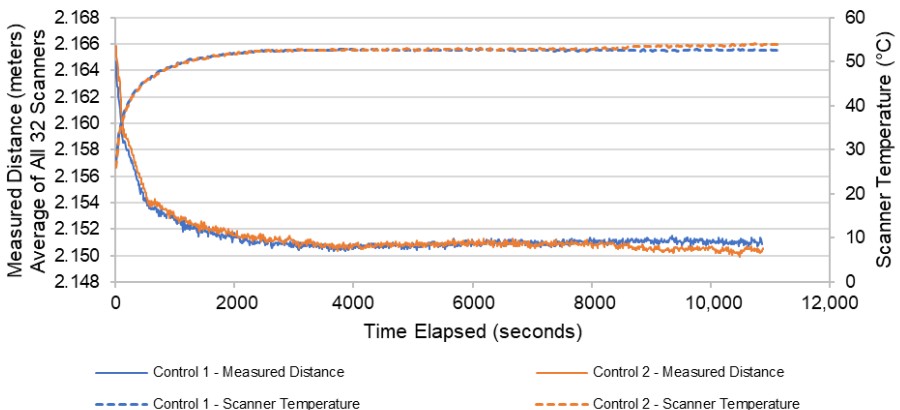

**Figure 16.** Velodyne® HDL–32E indoor experiment overall average measured distances (m) plotted over time elapsed with scanner internal temperature (°C).

At a glance, there appears to be a correlation between the two curves (average measured distance and internal scanner temperature) that is generally consistent between both experiments. With the exception of a minor deviation in temperature curve in Control 2, the experiment results were almost identical: while operating in an indoor office kept at 22 °C, the HDL–32E took approximately 30 min to reach a steady state of 52.6 °C, at which time the distance measurements stabilized. Both internal temperature and measured distance remained consistent after reaching this steady state.

To confirm the suspected correlation between the two parameters, the HDL–32E's internal temperature and overall average measured distance were plotted together and, using least squares to determine the slope and intercept coefficients, a linear trendline was fitted to the data (Figure 17). The resulting correlation coefficient (R) shows that the correlation between internal temperature and average measured distance is 0.9903 (very high), while coefficient of determination ($R^2$) of 0.9807 shows that 98.07% of the variation in the measured distance can be explained by the internal temperature. This informs an assumption that the HDL–32E's stability (or lack thereof) is highly correlated with its internal temperature, but this finding must be confirmed in survey-like conditions before it can be applied to UAS-lidar datasets.

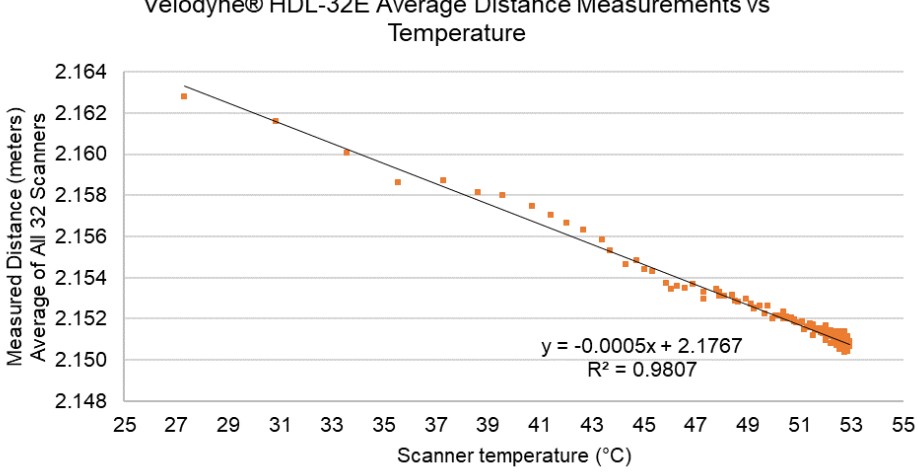

**Figure 17.** Linear relationship and correlation between Velodyne® HDL–32E internal temperature and overall average measured distance through the duration of indoor experiments. Equation for linear line of best fit and corresponding $R^2$ values are shown on the figure.

The relationship between internal scanner temperature and measured distance was further explored by plotting the average measured distances (m) for each individual channel with the scanner's internal temperature (°C). The resulting $R^2$ values for the overall scanner and each individual channel are shown in Figure 18, with the highest and lowest respectively shown in Figure 19. Of note, no individual channel has a higher $R^2$ value than the scanner as a whole (all scanner observations averaged together). Therefore, while it could be possible to generate and apply a linear regression for each individual scanner to potentially maximize the HDL–32E's accuracy, it may not be as effective as applying one linear regression to all individual scanners. As with the measures of precision, the coefficients of determination were not distributed evenly across the HDL–32E's central axis; there was no clear correlation between the linear measurement/temperature instability and offset angle of the channel. Additionally, there did not appear to be any relationship between channel precision and the calculated $R^2$ values per channel.

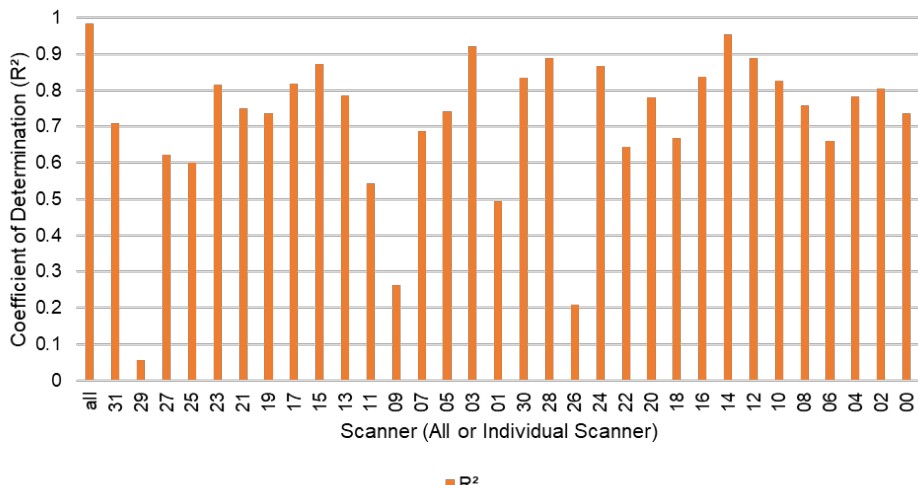

**Figure 18.** $R^2$ values for the linear trendlines and internal temperature/average measured distance for each of the Velodyne® HDL – 32E's individual channels.

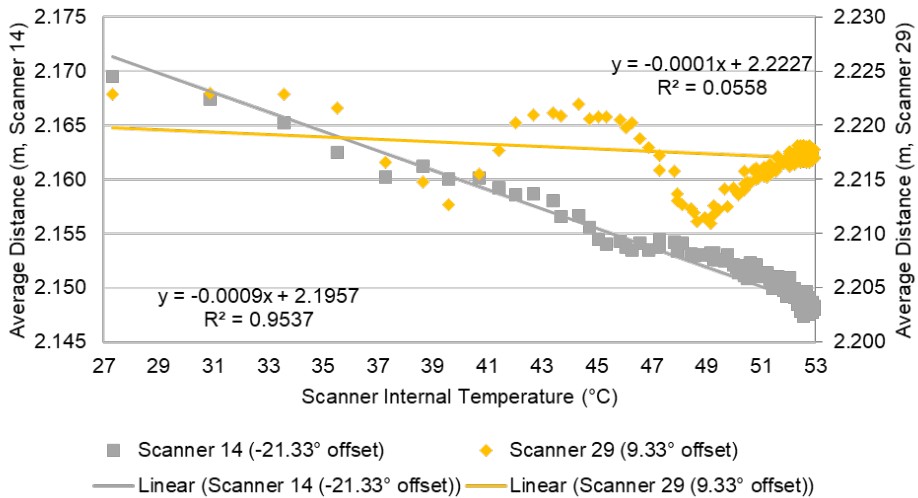

**Figure 19.** Linear relationship and correlation between Velodyne® HDL – 32E internal temperature and average measured distances for the individual channels with the highest (channel 14) and lowest (channel 29) $R^2$ through the duration of indoor experiments. Equation for linear line of best fit and corresponding $R^2$ values are shown on the figure.

### 3.1.2. Phase 2: Outdoor Experiments (Precision and Accuracy)

In the second phase, both the HDL–32E's precision and absolute accuracy were assessed through use of measurements collected from approximately 30 m under both day and night conditions, which were then compared to the observations of a Riegl® VZ-400 terrestrial laser scanner. Both day and night iterations of the absolute accuracy assessment exhibit range walks similar to those that were observed in all previous iterations. Because of this, the scanner's absolute accuracy varied over time; for approximately the first five minutes, the scanner's average measurements were too "long" (greater than the "true" distance), and after that point they were (generally) shorter than the calculated "true" distance.

Figure 20 shows the observed (HDL–32E) and "true" (VZ-400) measurement curves plotted with scanner internal temperature for the average distance between the target surface and scanner. These curves show that there was a loss of precision when day conditions are introduced to the scan, which in turn undermine the overall accuracy of the scanner. Additionally, while the average distance curves again exhibited an apparent

correlation with scanner temperature, there were anomalies in the day scan's line indicating that additional factors are influencing the scanner's stability.

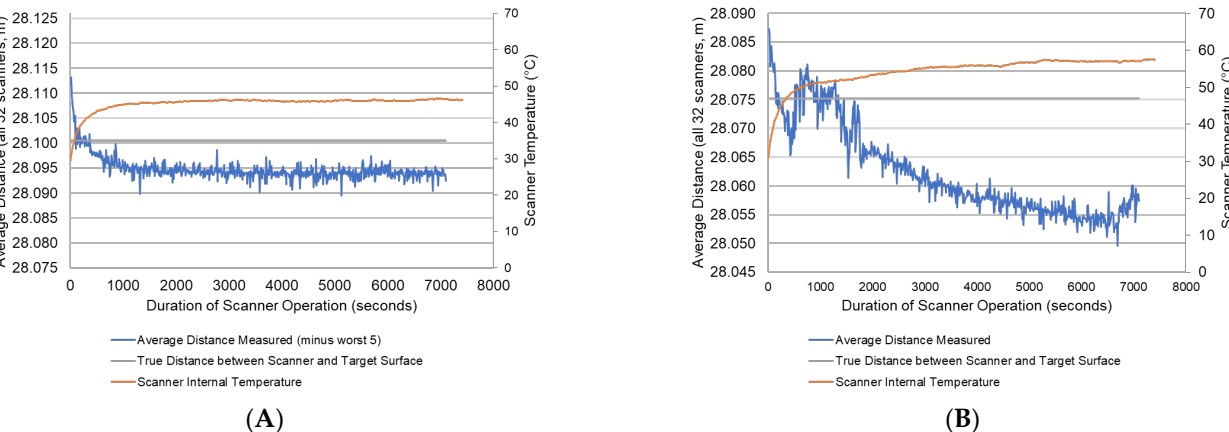

**Figure 20.** Average measured distance of all 32 individual channels versus internal scanner temperature and "true" distance value for both (**A**) night and (**B**) day iterations of the Velodyne® HDL–32E absolute accuracy experiment.

Between the night and day iterations (Figure 21A,B, respectively), there were several trends in scanner accuracy that may inform conclusions as to its optimization. the scanner's performance appears to optimize when the scan angle is closest to nadir. The full tables of calculated error (m) per individual scanner, per subsections of target surface for both the night and day experiments are available in Appendix A.

While offset angle did not appear to have a significant impact on scanner precision, the accuracy assessment experiments demonstrated that offset angle appeared to have a potentially significant impact on scanner accuracy. While individual scanner error is generally well within 0.015 m for most channels with offset angles less than 15°, individual channels with greater offset angles tend to exhibit greater error across all scan surfaces. With the exception of channel 4 during the day iteration, there were few examples of individual channels with offset angles greater than 15° with residuals that were regularly and significantly below the overall scanner RMSE.

Finally, while the individual channels that were the most accurate could vary between night and day iterations and even target surface subsections, the least accurate individual channels were much more consistent across experiment iterations. Channels 0, 2, 6, 8, and 10 showed the highest RMSE for all garage walls combined, and for all but one individual garage wall (channel 14 has an error of −0.037 m for the day scan of garage wall 4). This implies that scanner accuracy could be improved by simply removing these channels from the data—a course of action that is possible with both Velodyne® and Phoenix Lidar Systems® software. While doing so may not necessarily improve the HDL–32E's stability (channels 9, 19, 26, and 29 were the least precise during the night iteration and channels 11, 13, and 15 were least precise during the day iteration), the potential improvements to RMSE could justify the loss of collected data.

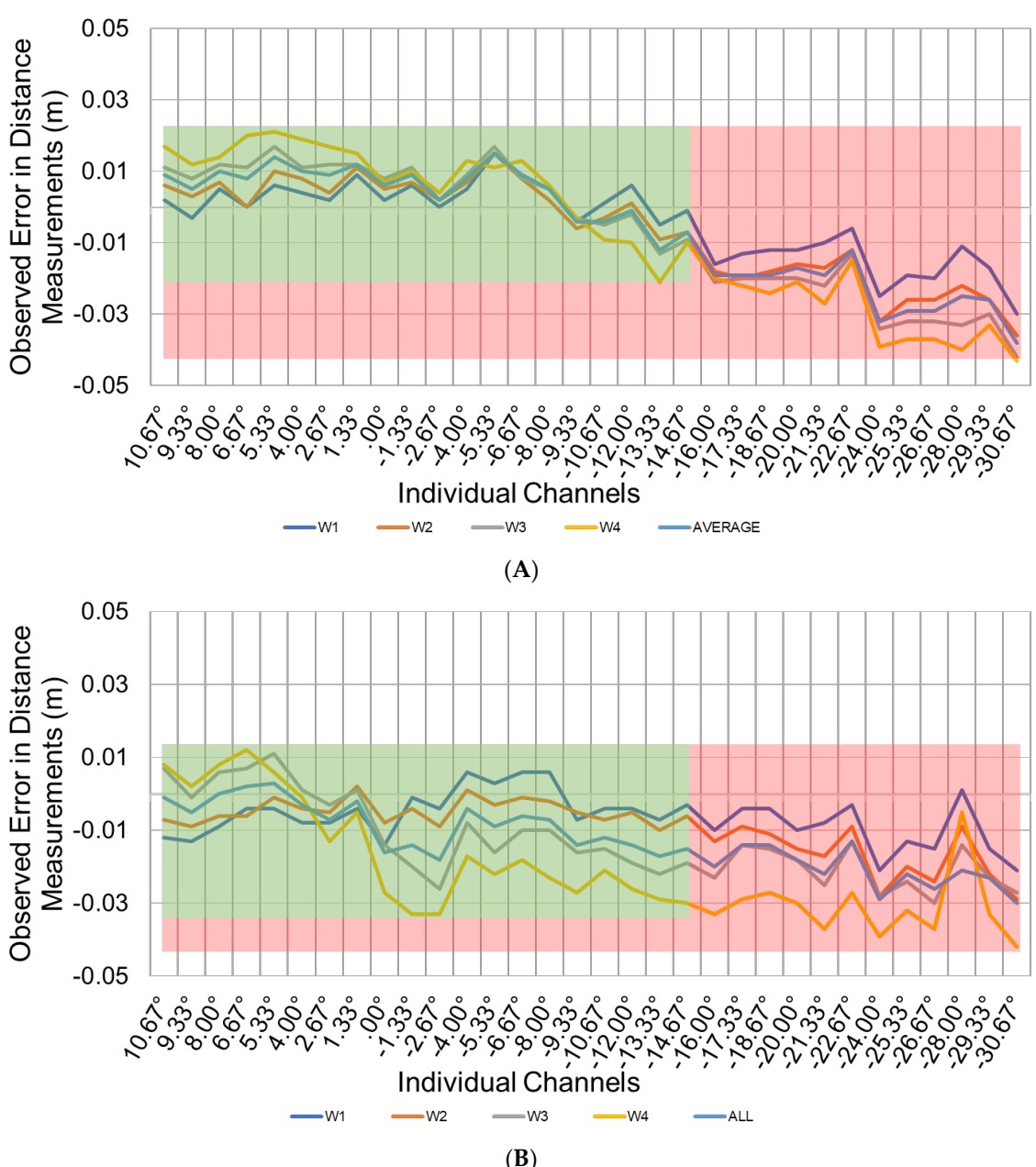

**Figure 21.** Calculated error (m) per individual channel, per subsections of target surface in Velodyne® HDL–32E during a 30 m (**A**) night experiment and a (**B**) day experiment. The green box highlights the reduced error observed when individual channels with offset angles greater than 15° are removed from the results, with the red box encompassing the total observed error by all individual scanners.

As expected, RMSE in the HDL–32E's observations was not consistent over time. Across the two hours of data collected, the RMSE was shown to generally increase from the beginning to the end of the experiments. This is in line with the results shown in Figure 20; while the scanner's observations could start relatively close to the "true" distance between scanner and target surface, it stabilized at a distance measurement that was potentially centimeters off of truth. This trend is further captured in Figure 22, where the RMSE of each observation file of the HDL–32E is plotted over the time elapsed in the experiment. The fact that the RMSE was generally stable while the scanner's measurements were stable implies that a correction could be more easily applied to reduce this error and increase the HDL–32E's accuracy. This RMSE is further broken down by time and target surface subsection in Appendix B.

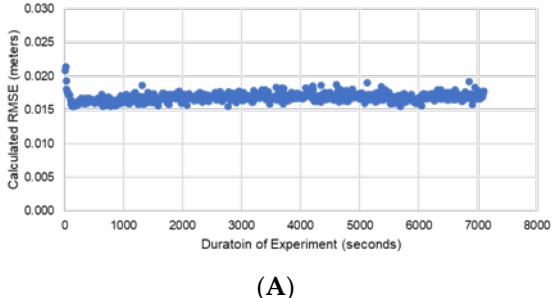

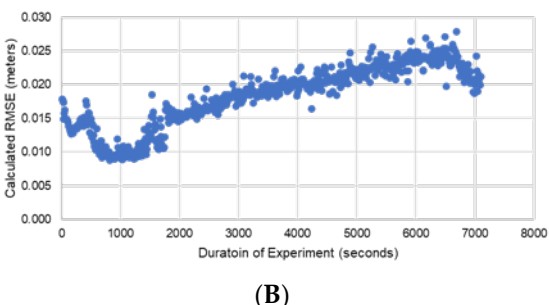

**Figure 22.** RMSE (m) per Velodyne® HDL–32E observation file for both (**A**) night and (**B**) day iterations of absolute accuracy experiments.

With the absolute accuracy of the scanner determined, it was possible to compare experiment parameters to observations and identify any potential impacts made by the environment. Again, three environmental conditions were used for this analysis: air temperature, atmospheric pressure, and solar radiation, shown in Figure 23.

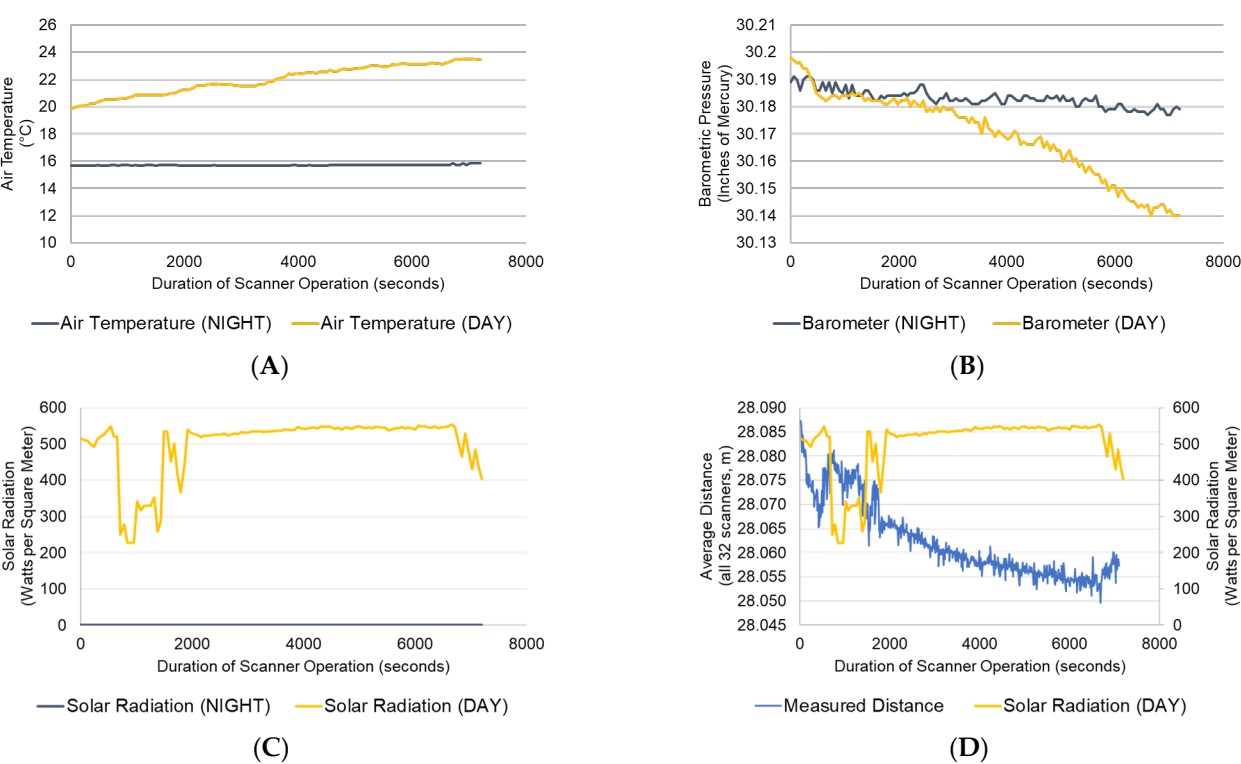

**Figure 23.** Weather conditions during Velodyne® HDL–32E night and day absolute accuracy experiments: (**A**) air temperature (°C), (**B**) barometric pressure (inches of mercury), and (**C**) solar radiation (watts per square meter) at study site. (**D**) Comparison between solar radiation and measured distance.

The higher air temperature of the day iteration versus the night iteration was expected (Figure 23A), and was associated with the higher scanner temperature of the day iteration shown in Figure 20A. The solar radiation results identify a phenomenon where a loss of stability in measured solar radiation coincided with a loss of scanner measurement stability. Figure 23D shows that where there was inconsistent solar radiation, there were inconsistent measurements made by the HDL–32E.

This loss of stability correlated with an experiment factor other than internal scanner temperature potentially undermines the observed correlation between scanner temperature and measured distance. This theory is confirmed in Figure 24, where the $R^2$ between

scanner internal temperature and measured distance increases from 0.7305 to a more significant 0.8650 with the observations affected by solar radiation removed.

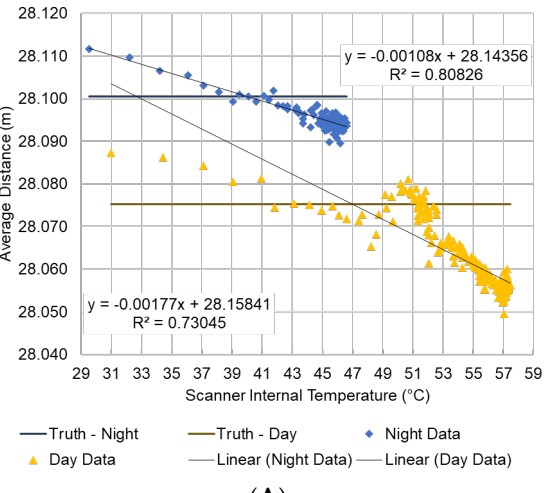
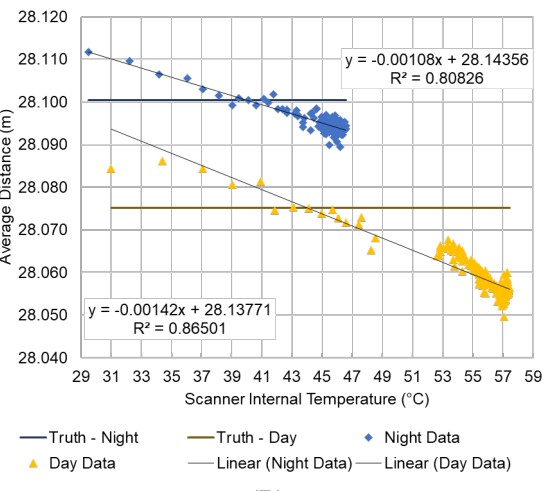

(**A**)　　　　　　　　　　　　　　　　　　　(**B**)

**Figure 24.** Linear relationship between Velodyne® HDL–32E average distance measurements and internal temperature for both night and day experiments, (**A**) with and (**B**) without the interference caused by solar radiation, with average truth values plotted for each experiment.

### 3.2. Livox® Mid–40

Following the same series of indoor test protocols as the Velodyne® HDL–32E, the Livox® Mid–40 was subjected to a series of indoor experiments to establish baseline of scanner behavior under controlled conditions. Because the Mid–40's field of view is limited to $\pm 19.2°$ in all directions from the normal of the scanner face, no restriction of scan angle was applied and the entire field of view was included in the analysis.

#### 3.2.1. Phase 1: Indoor Experiments (Precision)

Taking the average distance measured across the field of view per epoch and plotting it over the duration of the experiment, it was possible to identify overarching trends in the scanner's distance measurements. Figure 25 illustrates the range walk and autocorrection exhibited by the Mid–40, with an initial increase in measured distance of just under 0.020 m followed by a stabilization in both range and standard deviation after approximately 2000 s. Breaking the field of view down in to a 100 × 100 cell array, it is possible to visualize the trends in instability by both vertical and horizontal scan angle. The colorized raster is shown in Figure 26, and highlights, by percentile, the range of average measurements recorded per cell over time. The raster of calculated ranges shows that the across the entire field of view, the average range of distance measurements across the three hours observed was slightly higher than the change in measured distance found in the first 2000 s of the experiment (0.021 m versus the initial 0.017 m increase). This implies that, even after the scanner reached a relatively stable state, there was still a degree of instability in its measurements.

As the Mid–40 does not contain the function of internal scanner temperature measurement, it was not possible to determine what (if any) degree of correlation existed between the observed range walk and the operating temperature of the laser.

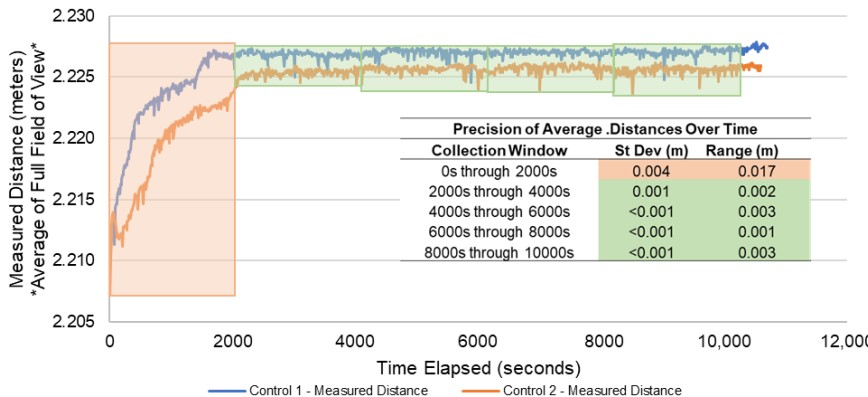

**Figure 25.** Average measured distance of the Livox® Mid–40's field of view over time during indoor experiments.

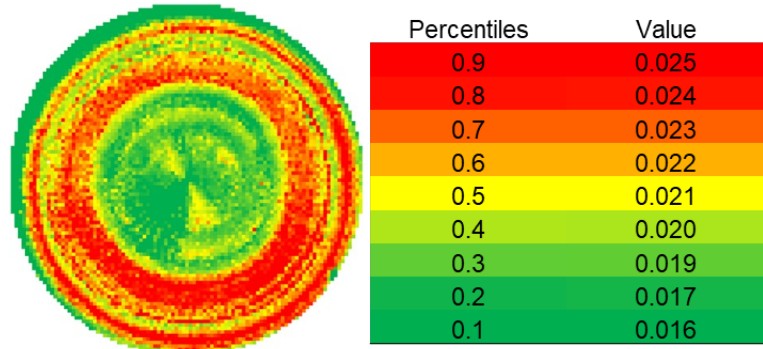

**Figure 26.** Raster of Livox® Mid–40 field of view, colorized by range of distance measurements per cell over duration of three-hour indoor experiments.

The field of view raster in Figure 26 illustrates a trend in the scanner's precision that is mentioned in [8], where a "ripple" effect is observed and reported. Rather than presenting a strong correlation between beam incident angle and precision, the pattern in precision change appears to demonstrate a loose correlation with the distribution on point density across the point cloud, at least in the center of the scan area (Figure 27). Because the rosette-style scan pattern cannot guarantee an even distribution of points across the target surface, there is a possibility that there is a corresponding loss of precision in range of measurements where there are consistently fewer observations (where the rosette has the least overlap).

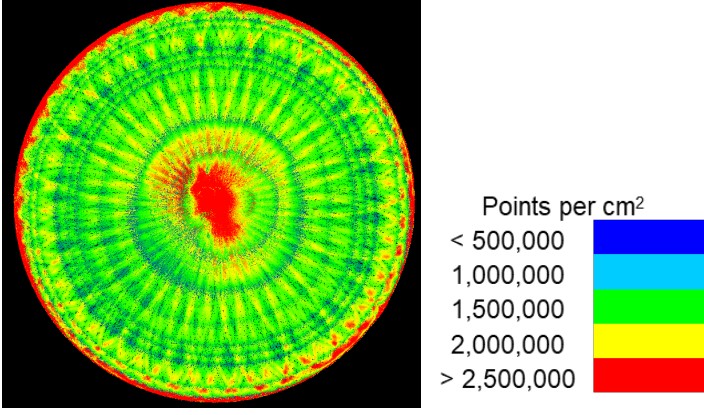

**Figure 27.** Livox® Mid–40 indoor point cloud (subset of 10 observation files), colorized by surface point density (points per cm).

### 3.2.2. Phase 2: Outdoor Experiments (Precision and Accuracy)

To determine which trends observed in the indoor experiments carry over to survey-like conditions, a precision assessment was conducted at a scanner distance of approximately 30 m. As some obstructions blocked portions of the Mid–40's field of view (Figure 28), the observations were limited to the four planar subsections of the target surface as indicated by Figure 8.

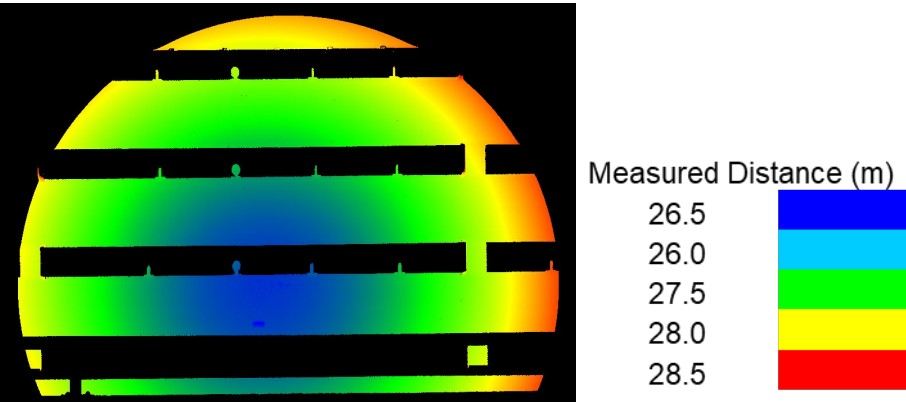

**Figure 28.** Raw observations of the Livox® Mid–40 during outdoor experiments, colored by measured distance from scanner origin.

The final phase of the Mid–40 assessment added the observations of the Riegl® VZ-400 to the outdoor experiments, which were repeated in both night and day conditions. The only significant change to workflow was the addition of processing the differences between observed measurements (from the Mid–40) and truth measurements (from the Riegl® VZ-400).

As shown in Figure 29, when the average measured distance per epoch of the Mid–40 is plotted over time, there is no significant difference between night and day experiments in general line shape. Both day and night experiments had a series of apparently randomly distributed outliers (Figure 29A), which were removed when filtered out using a 60-s running average low-pass style filter (Figure 29B). Both experiments had a drop in average measurements of <1 cm over the duration of the experiments, matching the results from the Phase 2 outdoor precision assessment.

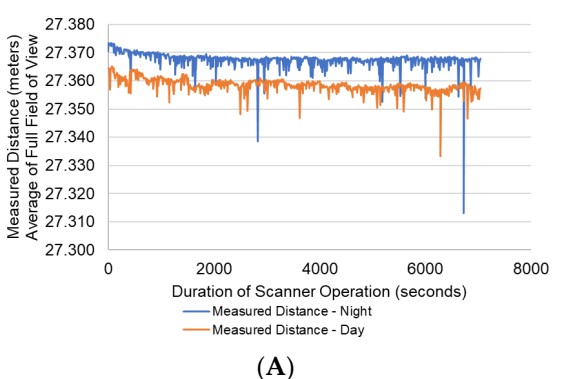

(**A**)

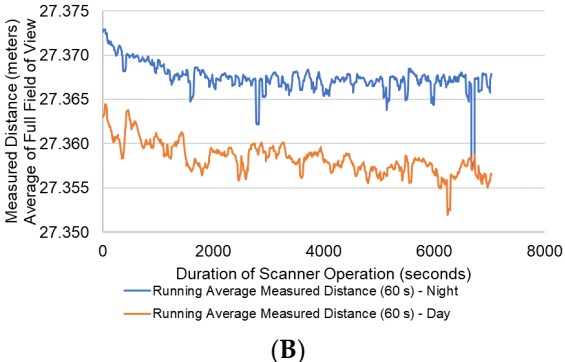

(**B**)

**Figure 29.** (**A**) Average measured distance and (**B**) running average measured distance per 60 s of the Livox® Mid–40's field of view over time during outdoor accuracy assessments.

Of note in the outdoor experiments was the observed range walk of a lesser magnitude and in the opposite direction compared to the indoor experiments. While it is possible that this change in direction is dictated by the operating temperature, it is difficult to establish a correlation between the two. The study area temperature for indoor experiments was an

average of 22.2 °C while the study area temperature for this outdoor experiment was an average of 14.2 °C. The calculated $R^2$ between the precision (range of measurements) and incident angle was found to be minimal with a value of 0.16.

When the range of measurements across the field of view are shown in a colorized raster (Figure 30), there is no significant difference in the Mid–40's distribution of precision between day and night experiments. Additionally, the "ripple" effect observed in the indoor experiment was lost at increased distance; instead of point density having an apparent correlation with instability, the incident angle of the beam appeared to have a moderate to strong correlation with measurement precision, confirming findings of [8].

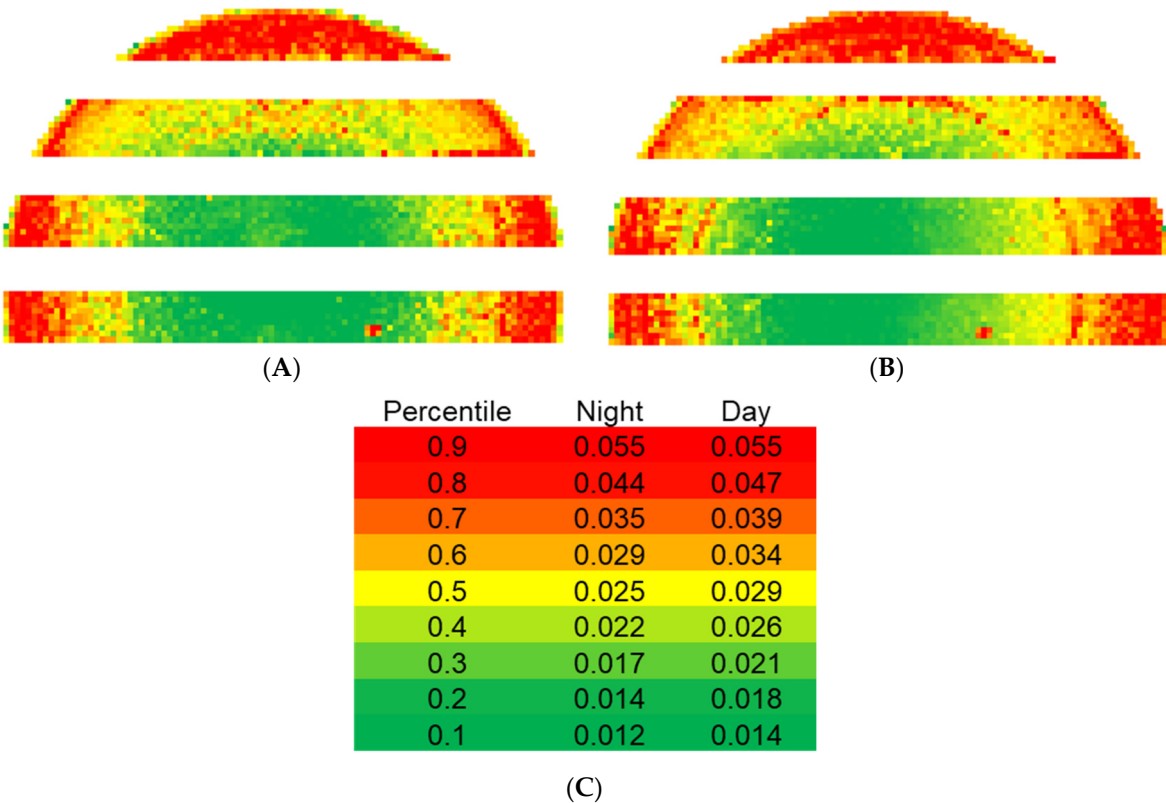

| Percentile | Night | Day |
|------------|-------|-------|
| 0.9 | 0.055 | 0.055 |
| 0.8 | 0.044 | 0.047 |
| 0.7 | 0.035 | 0.039 |
| 0.6 | 0.029 | 0.034 |
| 0.5 | 0.025 | 0.029 |
| 0.4 | 0.022 | 0.026 |
| 0.3 | 0.017 | 0.021 |
| 0.2 | 0.014 | 0.018 |
| 0.1 | 0.012 | 0.014 |

(C)

**Figure 30.** Range of measurements for the Livox® Mid–40 during (**A**) night and (**B**) day experiments, (**C**) colored by percentile.

In order to collect "truth" measurements, the points on the target surface as observed by the Mid–40 must be identified in the VZ-400's point cloud. This was accomplished using the alignment targets placed on the target surface observed by both scanners (Figure 31), yielding a sub-centimeter RMSE for both night and day alignments. The "true" distance between each point and the Mid–40's origin was assigned as a value to each identified point in the Mid–40's field of view.

Subtracting the "true" distance between point and Mid–40 origin for each point yielded the residual arrays shown in Figure 32, colorized by percentile. While there was minimal difference between night and day experiment precision, the day experiment had significantly more error in the middle-left portion of the field of view (−0.005 m for night, −0.013 m during the day). However, the average error for both experiments had a difference of less than a centimeter, with the majority of error in both experiments being in the negative (measured distances are shorter than "true" distances). The distribution of the error was also unique to this measure of scanner performance; there was no clear correlation between error and incident angle or point density.

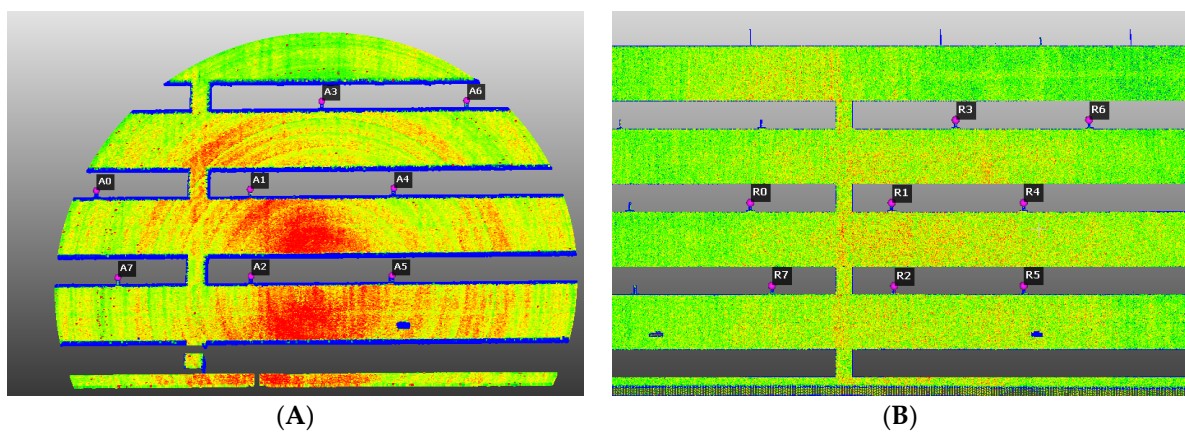

**Figure 31.** (**A**) Livox® Mid–40 and (**B**) Riegl® VZ-400 point clouds generated from observations of the target surface. CloudCompare [21] was used to register the two datasets using the targets towards extraction of conjugate points for measured distance comparison (RMSE of 0.007 m, and 0.005 m for night and day target registrations, respectively).

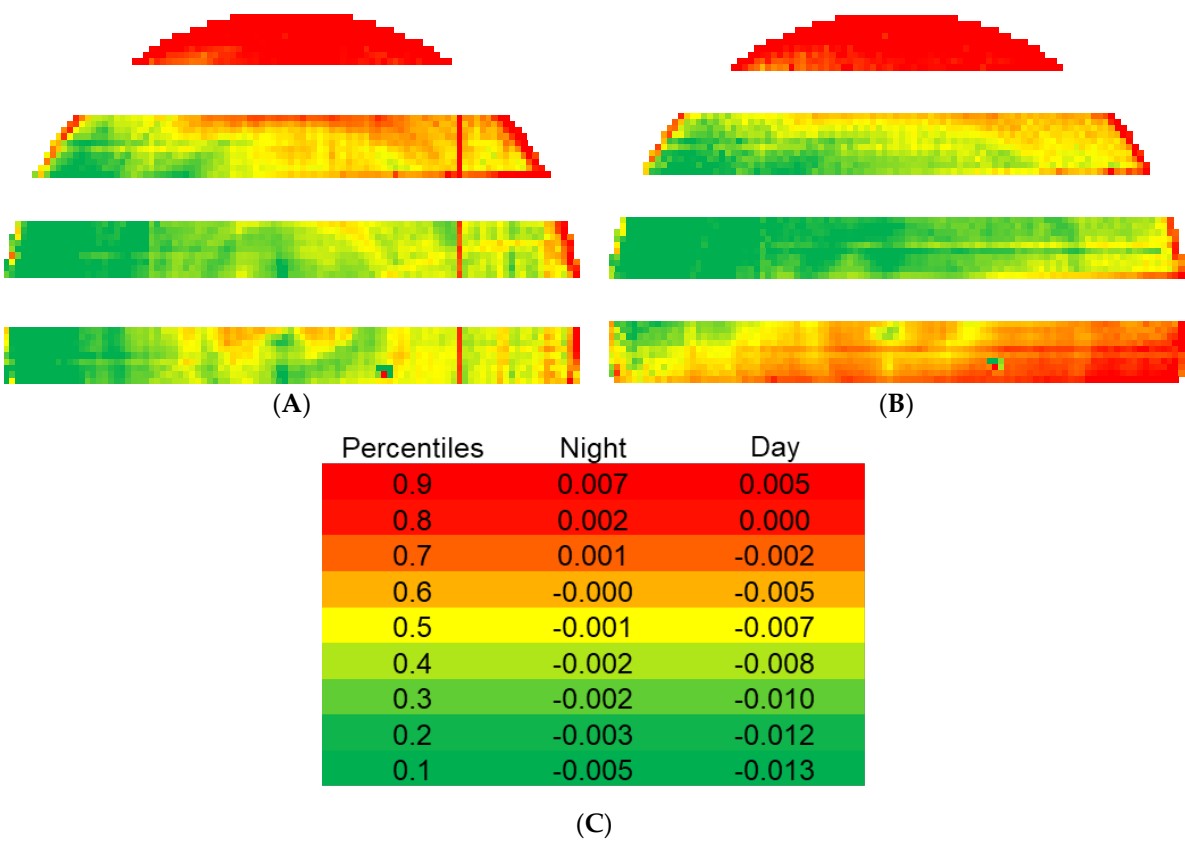

**Figure 32.** Livox® Mid–40 distribution of error in (**A**) night and (**B**) day experiments, (**C**) colored by percentile.

Calculating the RMSE for each experiment showed both night and day experiments had like levels of accuracy, with the RMSE of the night experiment at 0.004 m and the day at 0.009 m.

Parameter Correlations

As there was no clear significant difference between the results from night and day experiments, and there was likely no impact from observed experiment conditions. This observation was consistent for atmospheric pressure, solar radiation, and air temperature

(Figure 33). Additionally, the apparent loss of range walk observed between indoor and outdoor experiments did not appear to be related to experiment temperature, as the indoor experiments were conducted at an average of approximately 22.2 °C, and the accuracy assessment experiments were conducted at an average of 15.8 °C and 24.1 °C respectively.

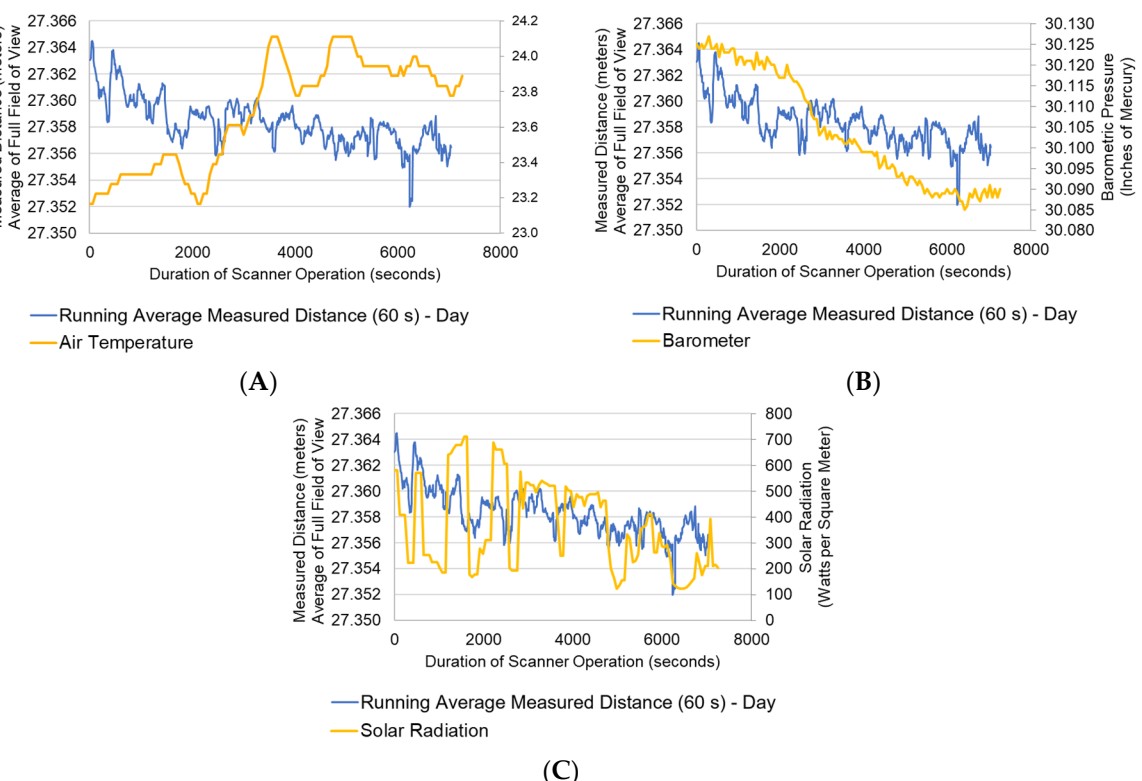

**Figure 33.** Livox® Mid–40 distance measurements (running average, per 60 s) and ongoing weather conditions (**A**) air temperature, (**B**) barometric pressure, and (**C**) solar radiation during day experiment.

The calculated $R^2$ between error and incident angle was found to be minimal at less than 0.1 for both and night experiments.

## 4. Discussion

### 4.1. Scanner Performance

#### 4.1.1. Temporal Instability

Both the HDL–32E and Mid–40 exhibited a degree of temporal instability and autocorrelation in their indoor control experiments (Figure 34). At a standoff distance of approximately 2 m, each experiment had a range walk of just under 2 cm until the scanner reached a state of relative stability after 30 min of operation. This trend was repeated for the HDL–32E during all outdoor experiments, but was lost for the Mid–40 once standoff between scanner and target surface was increased.

These findings confirm that is there a temporal bias in both scanners, though this bias is not uniform between the two scanners. The HDL–32E's range walk being highly correlated with its internal temperature further suggests that this instability in measured distance is the result of the heat generated by its continued operation. As its operating temperature cannot be directly measured in the Mid–40's current configuration, a like finding of correlation could not be confirmed in the Mid–40. However, given that both scanners have moving components, operate within similar voltage ranges, and demonstrate a range walk under like indoor conditions, it is possible that the Mid–40 is also subject to a similar internal temperature-correlated distance bias.

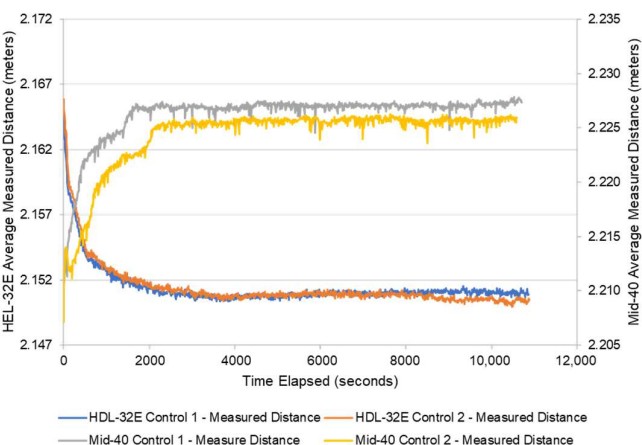

**Figure 34.** Temporal Stability of Velodyne® HDL–32E and Livox® Mid–40 during indoor experiments.

The loss of observed range walk at increased scanner standoff distance for the Mid–40 may be attributed to the minimal calibration of the scanner. The HDL–32E used in this study has been deliberately calibrated by the distributor, Phoenix Lidar Systems®, for high accuracy mobile mapping applications, whereas the Mid–40 only has the manufacturer's factory calibration. While the effectiveness of calibration methods is outside the scope of this study, the overall increase of noise visible in the Mid–40's point cloud at study distance versus the HDL–32E (and the HDL–32E's general lack of significant outlier measurements) implies that Mid–40 lacks the same capabilities as the HDL–32E to compensate for potential variability in its observations. This lack of calibration may have removed any impact of range walk in the Mid–40's outdoor experiments, but instrument bias being lost due to an increase in noise is hardly a viable solution to solving for scanner instability.

Regardless of why the range walk exists, the fact that it can be replicated under both controlled and survey conditions implies that it is an active source of error in the subject scanners, and must therefore be accounted for when executing surveys, topographic studies, or other high accuracy lidar applications.

4.1.2. Precision

The precision (repeatability) of the distance measurements made by the HDL–32E and Mid–40 were shown to have a moderate correlation with two separate scan parameters: the duration of scanner operation and the angle of beam emission relative to the scanner's X-axis.

When range walk could be observed, the precision of both the HDL–32E and the Mid–40 had a direct relationship to the duration of their operation; the longer the scanner operates, the more precise its measurements became. During the first 30 min of use, the range walk resulted in a 66% and 75% reduction in standard deviations and 93% and 88% reduction in range of average measurements for the HDL–32E and Mid–40, respectively (Figure 35). A like trend was observed in the HDL–32E during outdoor experiments, whereas the Mid–40 exhibited overall higher calculated standard deviation and range in average measurements across the duration of the experiment.

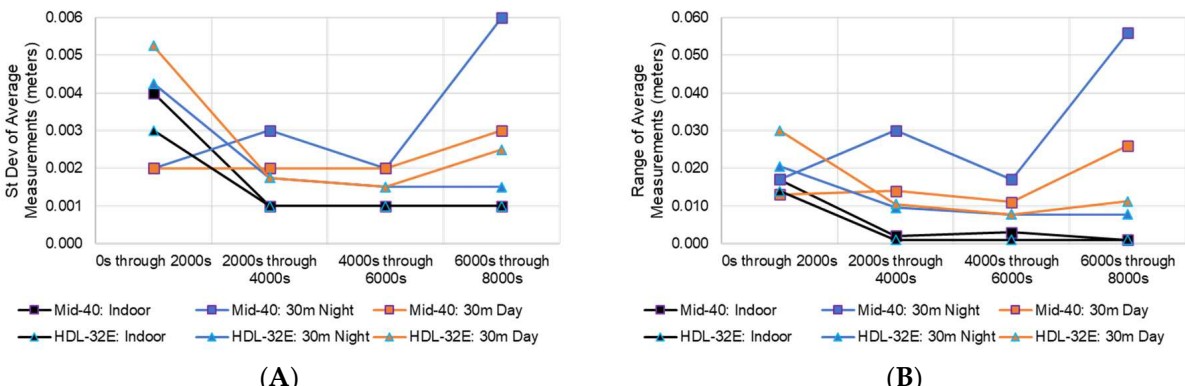

**Figure 35.** Calculated (**A**) standard deviation and (**B**) range of Velodyne® HDL–32E and Livox® Mid–40 over duration for all experiments.

While range walk was shown to impact scan precision over extended observation periods, the scan angle was found to impact measurement precision regardless of duration of scanner operation; measurement precision generally decreased as scan angle increases from nadir (Figure 36). In the HDL–32E, the horizontal scan angle (about the scanner's Z-axis) consistently had a higher correlation with loss of precision than vertical offset angle (about the Y-axis) and beam incident angle (angle of beam relative to planar normal of target surface). In the Mid–40, this relationship was further complicated by the varying results from indoor to outdoor experiments. The results from indoor experiments indicate that precision is related to point density on target surface, which is a product of beam emission angle and the scanner's rosette-style scan pattern. In outdoor experiments, this point density-precision relationship was mostly lost in favor of moderate correlations with scan angle and beam incident angle.

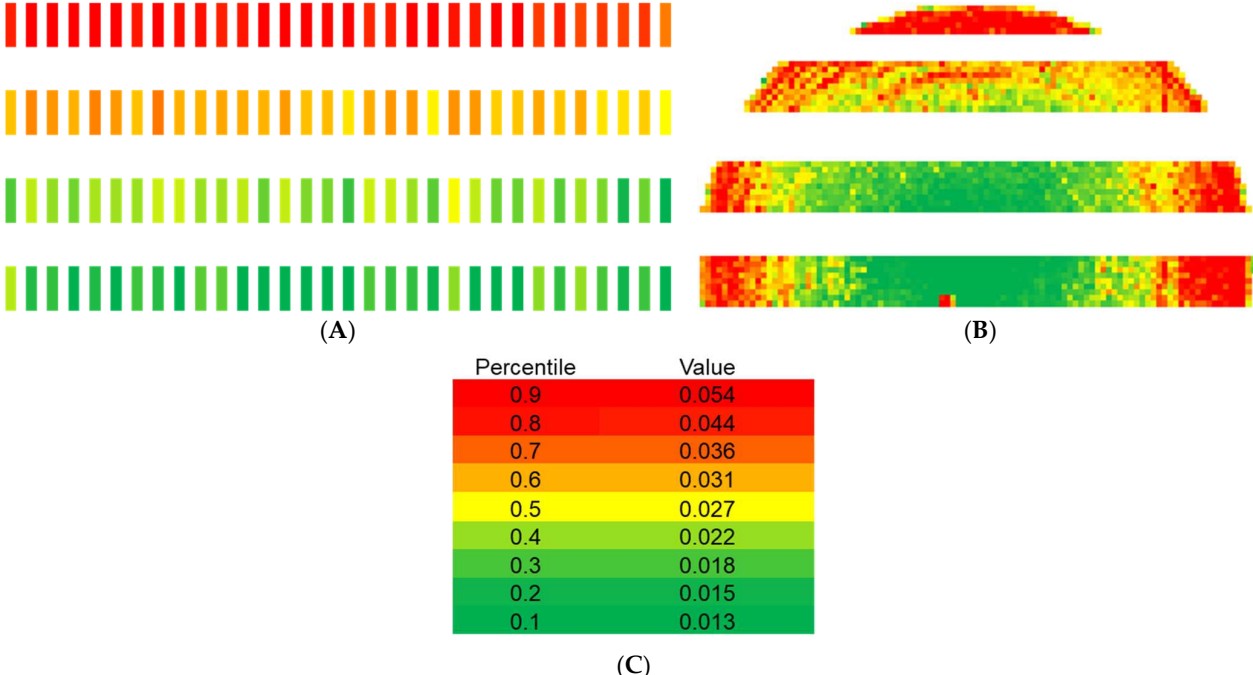

**Figure 36.** Distribution of precision (measurement range) for the (**A**) Velodyne® HDL–32E and (**B**) Livox® Mid–40 across their fields of view, (**C**) colorized by respective percentiles.

Given the findings regarding emission angle impact on scanner precision, it can be concluded that the most precise surveys were executed when the scanner (HDL–32E or

Mid–40) was observing targets near-nadir and after having been allowed to stabilize in its operating environment.

### 4.1.3. Accuracy

Scanner error, defined in this research by the residual between observed and "true" measurements, was shown to be influenced by the aforementioned factors that correlate to changes in temporal stability and precision. As any change in the scanner's measurements while held stationary changes the amount of observed error, this is a practical conclusion. However, this research shows that the presence of relatively high or low temporal stability or precision do not necessarily indicate the collection of the most accurate data.

As a whole, the HDL–32E exhibited an overall RMSE at or within the manufacturer's specifications (0.020 m) for both night and day experiments. In all accuracy assessments conducted with the HDL–32E, the initial overall measurements were regularly too far (positive error) before decreasing to and past the "true" distance and stabilizing at a consistent level measurement that was shorter than the "true" distance (negative error) (Figure 37). Furthermore, the relationship between each individual scanner's measurements and the "true" distance between scanner and observed point was not consistent across the array of individual laser/receiver pairs within the HDL–32E; there are individual scanners whose measurement curves, whether their measurements be low or high, never cross the "line" of "true" distance. The distribution of error across all 32 scanners was also not uniform; the largest error was found in scanners with offset angles greater than 15°, with the six scanners with the highest offset angle (scanners 0, 2, 4, 6, 8, and 10) having a level of error greater than 0.020 m.

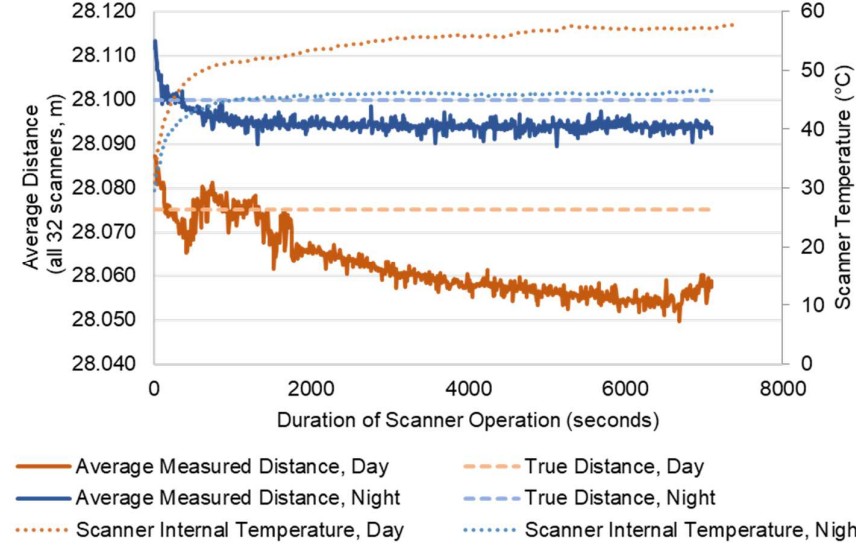

**Figure 37.** Accuracy of Velodyne® HDL–32E over time in night and day outdoor experiments. Shift in measured and "true" distance between night and day experiments is the result of minor adjustments made to scanner position after night iteration.

The Mid–40 also demonstrated an overall RMSE that was within manufacturer specifications (0.020 m) in its accuracy assessments. While not consistent in its spatial distribution or magnitude, the measured error (after filtering potential outliers in the data) stayed on average within a centimeter of the "true" distance. The temporal distribution of error was shown to be relatively stable with no clear trends regarding level of error and scan conditions over time. However, the amount of error during the day experiment was more than double that of the night, implying that the addition of daytime conditions limits the average scanner accuracy (Figure 38).

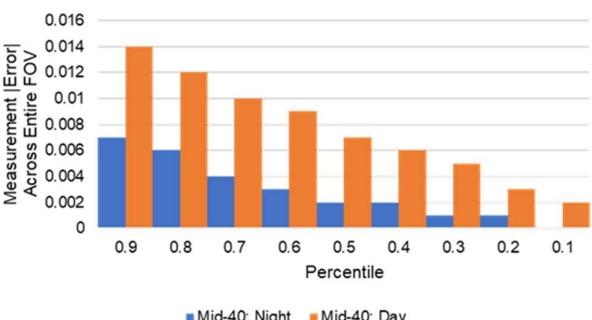

**Figure 38.** Amount of error observed in Livox® Mid–40 measurements in night and day scan conditions, by percentile.

### 4.2. Maximizing Accuracy

4.2.1. Solving for Error and Reducing RMSE

This study identified two main sources of error in the HDL–32E: scanner temperature and inaccurate individual laser emitter/detector pairs. Using the two successful accuracy assessments and the derived relationships between parameters presented in this study, three potential solutions were implemented to maximize scanner accuracy: removal of channels with high error, application of linear regressions developed from the same accuracy assessment (adjusting night measurements with regressions developed from the night accuracy assessment iteration, vice versa), and application of linear regressions developed from a separate, independent accuracy assessment (adjusting night dataset with day accuracy assessment results, vice versa). The resulting levels of RMSE before and after each of these techniques were implemented are shown in Figure 39.

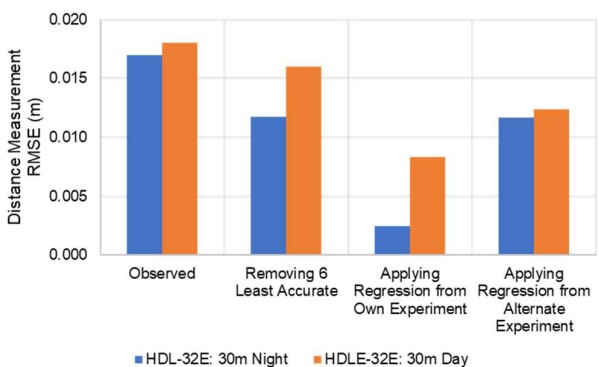

**Figure 39.** Application of error mitigation techniques on Velodyne® HDL–32E measurements.

The effectiveness of the linear regressions was further highlighted by applying them to each subsection of the target surface, extracted by the HDL–32E's scan azimuth. While accuracy and precision were shown to decrease as scan azimuth increases, Figure 40 shows that measurements of target subsections that have a higher measured RMSE than published standards could be corrected to within the published 0.020 m maximum.

Removal of the least accurate channels from the dataset, while a relatively simple solution, did not demonstrate a large enough impact on the HDL–32E's overall RMSE relative to other error mitigation techniques to justify loss of 20% or more of observed points. Instead, given the distribution of error over time, establishing and exploiting the relationship between error and scanner temperature demonstrated the most potential in maximizing measurement accuracy in a survey environment. Additionally, the net improvement in scanner RMSE using the regressions developed from alternate conditions (day regressions used for night data, night regressions for day data) implied that solutions developed under experiment conditions can be applied to observations from a variety of survey and mapping missions.

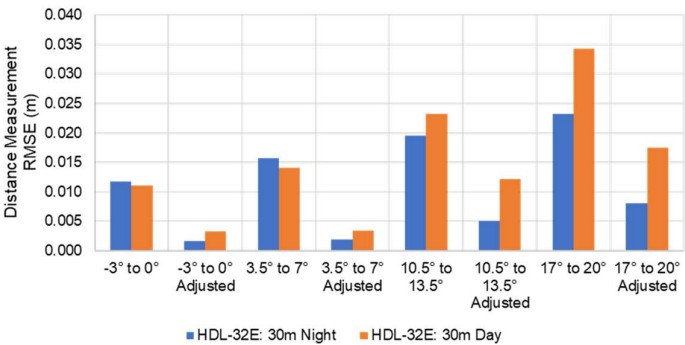

**Figure 40.** Applications of linear regressions on subsections of Velodyne® HDL–32E observations, split by scan azimuth.

As the Mid–40 has no observed parameters with as high of a correlation with measured error as internal temperature in the HDL–32E, the findings regarding its precision and accuracy do not lend themselves to as straightforward an approach as a linear regression. Applying the recommendations of [22] to filter the data in terms of reflectance and scan angle are effective at filtering noise and reducing error, and are recommended for all topographic uses of the scanner. However, it is likely that both a rigorous calibration and the installation of diagnostic tools (a thermometer) are required to identify and mitigate systemic errors in the scanner's distance measurements.

### 4.2.2. Impact of Scan Parameters and Conditions

Internal Scanner Temperature

While only observed in the HDL–32E, the internal scanner temperature is the parameter that has the highest observed correlation with average distance measurement with, in indoor experiments, repeated examples of a $R^2$ value in excess of 0.98. This correlation was observed in all experiments where the scanner internal temperature was recorded, with the lowest $R^2$ observed still being greater than 0.73. Though not consistent between individual scanners ($R^2$ ranging from 0.13 to 0.88), this study demonstrates that it is possible to build and implement linear regressions to reduce overall observed error. Given the range walk observed in the indoor tests with the Mid–40, it is implied that a similar relationship between internal scanner temperature and distance measurement may exist and could potentially be solved for in a like manner.

Solar Radiation

The disparity between night and day experiments for both the HDL–32E and Mid–40 in terms of precision and accuracy implies that the addition of sunlight on the target surface negatively impacts the overall performance of the scanners. Given that sunlight contains wavelengths of 905 nm (the wavelength used by both the HDL–32E and Mid–40), this is not unexpected. However, the fluctuations in observed solar radiation and the corresponding loss of precision observed in the HDL–32E implies that accounting for just the presence of sunlight during day experiments is insufficient for mitigating its effects. If collection at night is not possible due to safety concerns, local airspace restrictions (a lack of a waiver from the Federal Aviation Administration (FAA Part 107.29(b): operation at night) in the United States, for example), or survey requirements, it is recommended that consistent levels of solar radiation be prioritized during surveys to ensure consistent results.

Scan Angle

The HDL–32E and Mid–40 exhibited a moderate degree of correlation between laser scan angle and both their respective levels of precision and accuracy. Generally, as emission angle (and incident angle) increased, precision and accuracy decreased, with the lowest error and range of measurements being found near the instrument's nadir. However, no

correlation between performance and scan angle was found to meet or exceed that of other parameters (e.g., scanner temperature), and therefore developing a solution to account for identified error has not proven to be effective. Rather than retroactively solving for increased uncertainty at higher scan angles, it is recommended that data collection be executed with the highest feasible swath overlap to maximize the percentage of the study area observed near nadir. The resulting increase in redundant observations will likely mitigate errors from other sources present in the study area or instrument.

## 5. Conclusions

By conducting a deliberate assessment of the Velodyne® HDL–32E's and Livox® Mid–40's distance measurements in controlled and survey-like conditions, a holistic understanding of each scanner's spatial and temporal trends in accuracy and precision were developed. It was shown that, under the reported conditions and limitations, the HDL–32E and the Mid–40 each me and exceeded their published manufacturer's accuracy specifications, demonstrating potential for use in high-accuracy topographic surveys for a fraction the cost of their high-tier counterparts.

Each scanner exhibited a clear range walk of just under 0.020 m during the first 30 min of indoor conditions, which in the HDL–32E was found to be highly correlated with scanner internal temperature. This range walk and like correlation with temperature was found again in outdoor experiments with the HDL–32E, but was lost in the noise of the observations in the Mid–40. Allowing the scanners to operate for 30 min and stabilize in collection conditions prior to use effectively mitigated this instability.

Beyond the range walk, scanner precision and accuracy were found to be directly impacted by scan angle; as the angle of emission increased from the scanner's nadir, there was a moderately correlated loss of both precision and accuracy in both the HDL–32E and Mid–40. However, this angular offset was not equal in all axes, and the resulting loss of performance did not lend itself to a correction model. In order to mitigate this loss of precision and accuracy, it is recommended to condense flight lines for sUAS surveys and restrict scan angles to $\pm15°$ when observing generally flat or planar environments.

The presence of solar radiation on the target surface was found to decrease the precision and accuracy of scanner measurements to a minor degree, while fluctuations in solar radiation were found to be strongly correlated with potentially significant losses of precision. Limiting lidar collections to hours of night was found to reduce RMSE by up to 5% in the HDL–32E and 50% in the Mid–40.

By taking advantage of the relationship between scanner temperature and measured distance by each individual scanner of the HDL–32E, linear regressions were developed and used to adjust the reported distance measurements, resulting in a decrease in RMSE of over 88% when applied to like experiments. More importantly, a RMSE reduction of over 33% was found when regressions from one experiment were applied to the other, indicating that solutions developed in test conditions can be applied to datasets collected under a variety of conditions.

This research demonstrates a methodology to assess the temporal stability of lidar scanners that can be implemented by most lidar users. However, it is limited to just the two subject scanners. Future research should explore the performance of other low-cost lidar scanners from both Velodyne® and Livox®, as well as other manufacturers in the mobile mapping industry. Additionally, the impact of calibrations on the mapping of scanner stability (and its resulting mitigation) should be explored in future applications.

**Author Contributions:** Conceptualization: C.K. and B.W.; experiment methodology, C.K., O.C., B.W. and H.A.L.; software: C.K. and B.W.; formal analysis: C.K.; writing—original draft preparation: C.K.; writing—review and editing: B.W., A.A.-E., H.A.L. and O.C.; project administration: C.K., B.W. and A.A.-E. All authors have read and agreed to the published version of the manuscript.

**Funding:** This work was supported by the United States Department of Commerce—National Oceanic and Atmospheric Administration (NOAA) through The University of Southern Mississippi under the terms of Agreement No. NA18NOS400198, the USDA National Institute of Food and Agriculture, McIntire Stennis project FLA-FOR-005184.

**Data Availability Statement:** Data available on request from the primary author.

**Acknowledgments:** The authors would like to thank Ryan Brazeal for his subject matter expertise and advice regarding the use and study of the Livox® Mid–40. We also thank the University of Florida Police Department (UFPD) and the University of Florida's Transportation and Parking Services (TAPS) for their coordinated efforts to ensure successful data collections, and to Phoenix Lidar Systems® for technical support.

**Conflicts of Interest:** The authors declare no conflict of interest.

## Appendix A

While illustrated in Figure 21A,B, the calculated error per individual channel for both night and day experiments are provided to facilitate a more detailed understanding of the identified correlation between error and laser offset angle. In both iterations of the experiment, the observed error was found to be of a greater magnitude when the offset angle exceeded 15° (individual channels 22, 20, 18, 16, 14, 12, 10, 08, 06, 04, 02, and 00). This trend is repeated for all subsections of the target surface. Additionally, Tables A1 and A2 show the increased calculated error in distance measurements as the horizontal scan angle increases from near-perpendicular to the target surface (W1) to the highest observed scan angles (W4).

**Table A1.** Calculated error (m) per individual channel, per subsections of target surface in Velodyne® HDL–32E during 30 m, night experiment. Subsections are labeled W1 through W4, with W1 being the lowest observed and W4 being the highest.

| Channel | Offset | W1 | W2 | W3 | W4 | All |
|---|---|---|---|---|---|---|
| 31 | 10.67° | 0.002 | 0.006 | 0.011 | 0.017 | 0.009 |
| 29 | 9.33° | −0.003 | 0.003 | 0.008 | 0.012 | 0.005 |
| 27 | 8.00° | 0.005 | 0.007 | 0.012 | 0.014 | 0.010 |
| 25 | 6.67° | 0.000 | 0.000 | 0.011 | 0.020 | 0.008 |
| 23 | 5.33° | 0.006 | 0.010 | 0.017 | 0.021 | 0.014 |
| 21 | 4.00° | 0.004 | 0.008 | 0.011 | 0.019 | 0.010 |
| 19 | 2.67° | 0.002 | 0.004 | 0.012 | 0.017 | 0.009 |
| 17 | 1.33° | 0.009 | 0.011 | 0.012 | 0.015 | 0.012 |
| 15 | 0.00° | 0.002 | 0.005 | 0.008 | 0.007 | 0.006 |
| 13 | −1.33° | 0.006 | 0.007 | 0.011 | 0.010 | 0.009 |
| 11 | −2.67° | 0.000 | 0.002 | 0.002 | 0.004 | 0.002 |
| 9 | −4.00° | 0.005 | 0.007 | 0.009 | 0.013 | 0.008 |
| 7 | −5.33° | 0.015 | 0.015 | 0.017 | 0.011 | 0.015 |
| 5 | −6.67° | 0.009 | 0.008 | 0.008 | 0.013 | 0.009 |
| 3 | −8.00° | 0.005 | 0.002 | 0.005 | 0.006 | 0.005 |
| 1 | −9.33° | −0.004 | −0.006 | −0.003 | −0.003 | −0.004 |
| 30 | −10.67° | 0.001 | −0.003 | −0.005 | −0.009 | −0.004 |
| 28 | −12.00° | 0.006 | 0.001 | −0.002 | −0.010 | −0.001 |
| 26 | −13.33° | −0.005 | −0.009 | −0.013 | −0.021 | −0.012 |
| 24 | −14.67° | −0.001 | −0.007 | −0.009 | −0.010 | −0.007 |
| 22 | −16.00° | −0.016 | −0.018 | −0.021 | −0.020 | −0.019 |
| 20 | −17.33° | −0.013 | −0.020 | −0.020 | −0.022 | −0.019 |
| 18 | −18.67° | −0.012 | −0.018 | −0.020 | −0.024 | −0.019 |
| 16 | −20.00° | −0.012 | −0.016 | −0.020 | −0.021 | −0.017 |
| 14 | −21.33° | −0.010 | −0.017 | −0.022 | −0.027 | −0.019 |
| 12 | −22.67° | −0.006 | −0.012 | −0.013 | −0.015 | −0.012 |
| 10 | −24.00° | −0.025 | −0.032 | −0.034 | −0.039 | −0.032 |
| 8 | −25.33° | −0.019 | −0.026 | −0.032 | −0.037 | −0.029 |

**Table A1.** *Cont.*

| Channel | Offset | W1 | W2 | W3 | W4 | All |
|---|---|---|---|---|---|---|
| 6 | −26.67° | −0.020 | −0.026 | −0.032 | −0.037 | −0.029 |
| 4 | −28.00° | −0.011 | −0.022 | −0.033 | −0.040 | −0.025 |
| 2 | −29.33° | −0.017 | −0.026 | −0.030 | −0.033 | −0.026 |
| 0 | −30.67° | −0.030 | −0.036 | −0.042 | −0.043 | −0.038 |
| Overall Average Error | | −0.004 | −0.006 | −0.006 | −0.007 | −0.005 |
| RMSE (m) | | 0.011 | 0.015 | 0.019 | 0.022 | 0.018 |

**Table A2.** Calculated error (m) per individual channel, per subsection of target surface in Velodyne® HDL–32E during 30 m, day experiment. Subsections are labeled W1 through W4, with W1 being the lowest observed and W4 being the highest.

| Channel | Offset | W1 | W2 | W3 | W4 | All |
|---|---|---|---|---|---|---|
| 31 | 10.67° | −0.012 | −0.007 | 0.007 | 0.008 | −0.001 |
| 29 | 9.33° | −0.013 | −0.009 | −0.001 | 0.002 | −0.005 |
| 27 | 8.00° | −0.009 | −0.006 | 0.006 | 0.008 | 0.000 |
| 25 | 6.67° | −0.004 | −0.006 | 0.007 | 0.012 | 0.002 |
| 23 | 5.33° | −0.004 | −0.001 | 0.011 | 0.006 | 0.003 |
| 21 | 4.00° | −0.008 | −0.004 | 0.001 | −0.001 | −0.003 |
| 19 | 2.67° | −0.008 | −0.005 | −0.003 | −0.013 | −0.007 |
| 17 | 1.33° | −0.004 | 0.002 | 0.001 | −0.005 | −0.002 |
| 15 | 0.00° | −0.014 | −0.008 | −0.014 | −0.027 | −0.016 |
| 13 | −1.33° | −0.001 | −0.004 | −0.020 | −0.033 | −0.014 |
| 11 | −2.67° | −0.004 | −0.009 | −0.026 | −0.033 | −0.018 |
| 9 | −4.00° | 0.006 | 0.001 | −0.008 | −0.017 | −0.004 |
| 7 | −5.33° | 0.003 | −0.003 | −0.016 | −0.022 | −0.009 |
| 5 | −6.67° | 0.006 | −0.001 | −0.010 | −0.018 | −0.006 |
| 3 | −8.00° | 0.006 | −0.002 | −0.010 | −0.023 | −0.007 |
| 1 | −9.33° | −0.007 | −0.005 | −0.016 | −0.027 | −0.014 |
| 30 | −10.67° | −0.004 | −0.007 | −0.015 | −0.021 | −0.012 |
| 28 | −12.00° | −0.004 | −0.005 | −0.019 | −0.026 | −0.014 |
| 26 | −13.33° | −0.007 | −0.010 | −0.022 | −0.029 | −0.017 |
| 24 | −14.67° | −0.003 | −0.006 | −0.019 | −0.030 | −0.015 |
| 22 | −16.00° | −0.010 | −0.013 | −0.023 | −0.033 | −0.020 |
| 20 | −17.33° | −0.004 | −0.009 | −0.014 | −0.029 | −0.014 |
| 18 | −18.67° | −0.004 | −0.011 | −0.015 | −0.027 | −0.014 |
| 16 | −20.00° | −0.010 | −0.015 | −0.018 | −0.030 | −0.018 |
| 14 | −21.33° | −0.008 | −0.017 | −0.025 | −0.037 | −0.022 |
| 12 | −22.67° | −0.003 | −0.009 | −0.013 | −0.027 | −0.013 |
| 10 | −24.00° | −0.021 | −0.028 | −0.028 | −0.039 | −0.029 |
| 8 | −25.33° | −0.013 | −0.020 | −0.024 | −0.032 | −0.022 |
| 6 | −26.67° | −0.015 | −0.024 | −0.030 | −0.037 | −0.026 |
| 4 | −28.00° | 0.001 | −0.009 | −0.014 | −0.005 | −0.021 |
| 2 | −29.33° | −0.015 | −0.022 | −0.023 | −0.033 | −0.023 |
| 0 | −30.67° | −0.021 | −0.029 | −0.027 | −0.042 | −0.030 |
| Overall Average Error | | −0.007 | −0.009 | −0.013 | −0.021 | −0.013 |
| RMSE (m) | | 0.011 | 0.014 | 0.023 | 0.033 | 0.018 |

## Appendix B

In order to explore the HDL–32E's accuracy over time, the calculated RMSE can be broken down by garage wall, by collection window (every 2000 s or approximately 30 min, as was done for the precision analysis), shown in Tables A3 and A4.

**Table A3.** Calculated RMSE (m) per 2000 s of iteration, per subsection of target surface in Velodyne® HDL–32E during 30 m, night experiment. Subsections are labeled W1 through W4, with W1 being the lowest observed and W4 being the highest.

| Period | W1 | W2 | W3 | W4 | All |
|---|---|---|---|---|---|
| 0–2000 s | 0.012 | 0.015 | 0.019 | 0.023 | 0.017 |
| 2000–4000 s | 0.012 | 0.016 | 0.019 | 0.023 | 0.017 |
| 4000–6000 s | 0.012 | 0.016 | 0.020 | 0.023 | 0.017 |
| 6000–8000 s | 0.012 | 0.016 | 0.019 | 0.024 | 0.017 |
| All | 0.012 | 0.016 | 0.019 | 0.023 | 0.017 |

**Table A4.** Calculated RMSE (m) per 2000 s of iteration, per subsection of target surface in Velodyne® HDL–32E during 30 m, day experiment. Subsections are labeled W1 through W4, with W1 being the lowest observed and W4 being the highest.

| Period | W1 | W2 | W3 | W4 | All |
|---|---|---|---|---|---|
| 0–2000 s | 0.008 | 0.011 | 0.016 | 0.025 | 0.012 |
| 2000–4000 s | 0.011 | 0.013 | 0.022 | 0.034 | 0.018 |
| 4000–6000 s | 0.012 | 0.016 | 0.027 | 0.039 | 0.022 |
| 6000–8000 s | 0.014 | 0.018 | 0.027 | 0.038 | 0.023 |
| All | 0.011 | 0.014 | 0.023 | 0.033 | 0.018 |

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
