# Peer review of "Accuracy Assessment of Low-Cost Lidar Scanners: An Analysis of the Velodyne HDL–32E and Livox Mid–40’s Temporal Stability"

_remotesensing, doi:10.3390/rs14174220_

Round 1

Reviewer 1 Report

Dear authors,

I find the article interesting and beneficial to the knowledge about lidars used especially in mobile (and especially UAV) mapping systems. 

I have overall two comments:

1. the Livox lidar sensor is used in the low-cost UAV scanner DJI Zenmuse L1, so its test is very important. I would recommend adding this to the article, noting that tests of complete systems are also performed (e.g. here: Teppati Losè, L., Matrone, F., Chiabrando, F., Giulio Tonolo, F., Lingua, A., and Maschio, P.: NEW DEVELOPMENTS IN LIDAR UAS SURVEYS. PERFORMANCE ANALYSES AND VALIDATION OF THE DJI ZENMUSE L1, Int. Arch. Photogramm. Remote Sens. Spatial Inf. Sci., XLIII-B1-2022, 415-422, https://doi.org/10.5194/isprs-archives-XLIII-B1-2022-415-2022, 2022,

as well as others)

Indeed, there is a question of the significance of an accuracy improvement of the order of 1 cm - 2 cm compared to e.g. georeferencing error in higher values (because it is a mobile device, usually dependent on GNSS RTK coordinates and orientation, e.g. for the already mentioned DJI L1 the manufacturer states 10 cm and 5 cm (position and height)).  

Using the example of the accuracies achieved in the tests of the whole DJI L1 system, it can be shown that e.g. 2 cm is already a significant value, since even with the already mentioned DJI L1, accuracies in units of centimetres are achieved. 

2. In almost all figures there is doubled the description - in figure itself and in the figure's caption. This should be corrected.

Best Regards

Author Response

To the reviewer,

Before all else, thank you for your time and energy in providing your review for our manuscript. We understand your time is valuable and we deeply appreciate your feedback. Please see below for our responses to your comments.

Comment 1: The Livox lidar sensor is used in the low-cost UAV scanner DJI Zenmuse L1, so its test is very important. I would recommend adding this to the article, noting that tests of complete systems are also performed (e.g. here: Teppati Losè, L., Matrone, F., Chiabrando, F., Giulio Tonolo, F., Lingua, A., and Maschio, P.: NEW DEVELOPMENTS IN LIDAR UAS SURVEYS. PERFORMANCE ANALYSES AND VALIDATION OF THE DJI ZENMUSE L1, Int. Arch. Photogramm. Remote Sens. Spatial Inf. Sci., XLIII-B1-2022, 415-422, https://doi.org/10.5194/isprs-archives-XLIII-B1-2022-415-2022, 2022, as well as others).

Indeed, there is a question of the significance of an accuracy improvement of the order of 1 cm - 2 cm compared to e.g. georeferencing error in higher values (because it is a mobile device, usually dependent on GNSS RTK coordinates and orientation, e.g. for the already mentioned DJI L1 the manufacturer states 10 cm and 5 cm (position and height)). Using the example of the accuracies achieved in the tests of the whole DJI L1 system, it can be shown that e.g. 2 cm is already a significant value, since even with the already mentioned DJI L1, accuracies in units of centimetres are achieved. 

Response: We thank the reviewer for their valuable feedback. The mentioned text is an excellent resource, and it is particularly applicable to some of our ongoing research regarding comparisons of point clouds created by lidar and structure from motion. However, the lidar scanner Teppati Losè et al., present in their analysis (the Livox® Avia, subcomponent of the DJI Zenmuse L1) is a different model than the scanner tested in this manuscript (the Livox® Mid-40). We have included it in our literature review, as its findings regarding the Avia’s limitations in capturing objects defined by sharp edges are similar to findings of like studies with the Mid-40 (line 106).

We are pleased to be in agreement with the reviewer that improving scanner accuracy by centimeters can be significant, especially given the manufacturer’s accuracy specifications for several systems currently on the market. While our research does not address the propagation of scanner measurement error through a complete lidar scanning system during an airborne survey, we are looking forward to applying our findings to future datasets and validating our recommendations.

Comment 2: In almost all figures there is doubled the description - in figure itself and in the figure's caption. This should be corrected.

Response: As identified by the reviewer, redundant descriptions in tables and figures have been removed throughout the manuscript.  

Should additional clarification be needed or corrections required in our text, please do not hesitate to let us know.

Very respectfully,

Carter Kelly, Primary Author

Reviewer 2 Report

This is a thorough, in-depth, and sound research paper on the detailed experimental analysis and engineering evaluation of scanner errors and accuracy of two important low-cost commercial lidar mapping systems.   The experiments outlined are well documented, are complete, and show that the measurement error can be improved by up to 60% through the use of autocorrelation analysis of systemic errors.   The resultant measurement accuracy is then comparable to that of more costly lidar systems; this latter point could be added to the abstract to help the paper reach a larger audience.

Author Response

To the reviewer,

Before all else, thank you for your time and energy in providing your review for our manuscript. We understand your time is valuable and we deeply appreciate your feedback. Please see below for our responses to your comments.

Comment 1: This is a thorough, in-depth, and sound research paper on the detailed experimental analysis and engineering evaluation of scanner errors and accuracy of two important low-cost commercial lidar mapping systems. The experiments outlined are well documented, are complete, and show that the measurement error can be improved by up to 60% through the use of autocorrelation analysis of systemic errors. The resultant measurement accuracy is then comparable to that of more costly lidar systems; this latter point could be added to the abstract to help the paper reach a larger audience.

Response: We thank the reviewer for their valuable feedback and have added their recommended point regarding more expensive systems to our abstract (Line 31).

Should additional clarification be needed or corrections required in our text, please do not hesitate to let us know.

Very respectfully,

Carter Kelly, Primary Author

Reviewer 3 Report

The presentation of these experimentations is clear and may be useful for some readers. I would suggest some minor improvements :

- In each figure where a parameter is represented by a colour, please check that a table is provided to give the correspondance colour vs value of the parameter. This could help the reader.

- In Fig. 23, there is a discrepancy in the type of temperatures used (°F vs °C). And generally, for non US readers, it would be preferable to use only °C

Author Response

To the reviewer,

Before all else, thank you for your time and energy in providing your review for our manuscript. We understand your time is valuable and we deeply appreciate your feedback. Please see below for our responses to your comments.

Comment 1: In each figure where a parameter is represented by a colour, please check that a table is provided to give the correspondence colour vs value of the parameter. This could help the reader.

Response: We thank the reviewer for their feedback; where relevant parameters are represented by a color in figures, a corresponding table of color values has been added.  

Comment 2: In Fig. 23, there is a discrepancy in the type of temperatures used (°F vs °C). And generally, for non US readers, it would be preferable to use only °C

Response: Instances where there are discrepancies between units of temperature depicted versus described have been identified and fixed throughout the manuscript for a uniform °C.

Should additional clarification be needed or corrections required in our text, please do not hesitate to let us know.

Very respectfully,

Carter Kelly, Primary Author

Reviewer 4 Report

Dear Authors, congratulation for your work. I just put some notes inside the text but in general no specific observations

Author Response

To the reviewer,

Before all else, thank you for your time and energy in providing your review for our manuscript. We understand your time is valuable and we deeply appreciate your feedback. Please see below for our responses to your comments.

The presentation of these experimentations is clear and may be useful for some readers. I would suggest some minor improvements:

Comment 1: Line 236: “using Equation 1”

Response: Error has been identified and correction has been made (Line 240)

Comment 2: Line 342: Please, specify if you refer to range measurements or coordinates

Response: Upon reviewing the manufacturer specifications and the calibration sheets provided by Riegl©, we can confirm that the accuracy ratings are regarding distance measurements (Line 347).

Comment 3: Line 367: please use "estimate" instead of "calculate"

Response: Correction has been made (Line 371).

 Comment 4: Line 579: maybe "measurements"?

Response: We have adjusted our word choice to clarify our meaning (Line 594).

Should additional clarification be needed or corrections required in our text, please do not hesitate to let us know.

Very respectfully,

Carter Kelly, Primary Author

Reviewer 5 Report

This paper presents an analysis of range measurement instabilities in low cost lidar sensors. Modelling and error mitigation approaches are also reported. The paper is timely relevant and addresses gaps in current knowledge: the cause of temporal instability of the Velodyne; the performance of the Livox scanner. This is a very thorough investigation. The authors should be commended for the meticulous detail with which they have conducted and reported their experiments. Scanning the scanners themselves with another instrument is a clever idea to allow absolute accuracy assessment.

Listed below are a number of minor revisions to improve the paper. Most pertain to experiment details and presentation. The use of “scanner” to refer to an individual laser channel needs to be corrected.

Specific comments:

Line 198

What is the wall material?

Line 248

Change the temperature units to metric

Line 260-265

The details of manipulating CSV files are not important. Only the channel separation is of interest.

Figure 7

Add the colour scale for both A) and B)

Line 306

What is the parking garage material? Concrete?

Line 437

I find the scanner terminology confusing

“so it is possible to plot the observations of each of the 32 laser scanners within the HDL-32E”

Do the authors mean 32 lasers? The whole instrument is a scanner that comprises 32 individual lasers

Figure 19

These results are quite interesting and raise a couple questions:

1.      How repeatable is the temperature-distance correlation from one day to the next? If one calibrates the instrument using the regression parameters determined on one day, will they be relevant the next day? I see this is addressed later

2.      Both sets of data in this figure exhibit what look like periodic deviations from the linear trend, albeit with different amplitudes. Perhaps a higher-order model that captures this behaviour is relevant for each individual laser.

Figure 21

Instead of plotting error as a function of laser number, plotting as a function of offset angle would be better. The laser numbers could be added below.

Line 638

I think the explanation is only partly true, certainly in the centre. Perhaps plot precision as a function of density for a more clear visualization about the correlation.

Figure 27

Add colour scale

Author Response

To the reviewer,

Before all else, thank you for your time and energy in providing your review for our manuscript. We understand your time is valuable and we deeply appreciate your feedback. Please see below for our responses to your comments, and please note that the line number after each comment reflect the new line the correction can be found on.

Comment 1:  Line 198 - What is the wall material?

Response: The wall materials used through this study are painted drywall (indoor experiments) and exposed concrete (outdoor experiments). We have clarified the wall materials in text (lines 202, 252, and 311).  

Comment 2: Line 248 - Change the temperature units to metric

Response: As per the reviewer’s recommendation, we have changed all temperature units throughout this text to degrees Celsius.

Comment 3: Line 260-265 - The details of manipulating CSV files are not important. Only the channel separation is of interest.

Response: We thank the reviewer for their feedback. To properly highlight the channel separation portion of the workflow, we have re-constructed our explanation to remove references to file type and other miscellaneous details (Lines 264-269).

Comment 4: Figure 7 - Add the colour scale for both A) and B)

Response: As per the reviewer’s recommendations, color scales have been added to both figures to clarify depicted values.

Comment 5: Line 306 - What is the parking garage material? Concrete?

Response: The outdoor target surface is exposed concrete; as per the reviewer’s comment, we have clarified the target surface materials in text (Line 311).

Comment 6: Line 437 - I find the scanner terminology confusing: “so it is possible to plot the observations of each of the 32 laser scanners within the HDL-32E” Do the authors mean 32 lasers? The whole instrument is a scanner that comprises 32 individual lasers

Response: We thank the reviewer for identifying this issue. The line of text in question is in reference to the 32 individual laser channels of the HDL-32E. To avoid confusion, we have changed the text to read “32 channels within the HDL-32E”, and have reviewed the rest of the manuscript to ensure continuity in terminology (Line 441).

Comment 7: Figure 19 - These results are quite interesting and raise a couple questions:

  1. How repeatable is the temperature-distance correlation from one day to the next? If one calibrates the instrument using the regression parameters determined on one day, will they be relevant the next day? I see this is addressed later
  2. Both sets of data in this figure exhibit what look like periodic deviations from the linear trend, albeit with different amplitudes. Perhaps a higher-order model that captures this behaviour is relevant for each individual laser.

Response: 1. Excellent question. The indoor experiments resulted in a high degree of consistency across different iterations. This included both the scanner as a whole (Figure 16) and individual channels within the HDL-32E, and formed the basis for our conclusion that, under like (stable) conditions, the scanner has a consistent “range walk” that can be identified and potentially solved for. For the indoor experiments, however, there was no “truth” data to form the basis for a data correction, so use of the indoor regression parameters for adjustments is limited. As pointed out by the reviewer, this is addressed later in the manuscript when we have “truth” data for the outdoor experiments, and we find that, should scan conditions be similar, regression parameters can be used between different data collections.

  1. The reviewer is correct in pointing out that higher-order models exhibit a better fit (higher R2) for channels such as channel 29 shown in Figure 19. However, when we determined which order model resulted in the highest R2 per channel and applied them in later experiments where “truth” data was available, it was found that the reduction in measurement error was negligible.

Comment 8: Figure 21 - Instead of plotting error as a function of laser number, plotting as a function of offset angle would be better. The laser numbers could be added below.

Response: We thank the reviewer for their feedback and have adjusted the horizontal axis labels in Figure 21 to reflect their recommendation.

Comment 9: Line 638 - I think the explanation is only partly true, certainly in the centre. Perhaps plot precision as a function of density for a more clear visualization about the correlation.

Response: The correlation between point density and range of measurements identified in this section, as the reviewer rightfully point out, is relatively loose and does not fully explain the phenomena observed. Due to the modest nature of the potential correlation, plotting precision as a function of density fails of offer a more-clear visualization. Instead, we have re-phrased our explanation to reflect the moderate relationship between the two parameters (Lines 664-661).

Comment 10: Figure 27 - Add colour scale

Response: As per the reviewer’s recommendations, a color scale have been added to Figure 27 to clarify depicted values.

Should additional clarification be needed or corrections required in our text, please do not hesitate to let us know.

Very respectfully,

Carter Kelly, Primary Author